

# Gauging tensor networks with belief propagation

**Joseph Tindall⋆ and Matt Fishman**

Center for Computational Quantum Physics, Flatiron Institute,
New York, New York 10010, USA

⋆ jtindall@flatironinstitute.org

## Abstract

Effectively compressing and optimizing tensor networks requires reliable methods for fixing the latent degrees of freedom of the tensors, known as the gauge. Here we introduce a new algorithm for gauging tensor networks using belief propagation, a method that was originally formulated for performing statistical inference on graphical models and has recently found applications in tensor network algorithms. We show that this method is closely related to known tensor network gauging methods. It has the practical advantage, however, that existing belief propagation implementations can be repurposed for tensor network gauging, and that belief propagation is a very simple algorithm based on just tensor contractions so it can be easier to implement, optimize, and generalize. We present numerical evidence and scaling arguments that this algorithm is faster than existing gauging algorithms, demonstrating its usage on structured, unstructured, and infinite tensor networks. Additionally, we apply this method to improve the accuracy of the widely used simple update gate evolution algorithm.

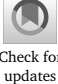

# 1 Introduction

Tensor networks are low-rank approximations of high-order, potentially infinite-order, tensors as products of smaller tensors [1–13]. Tensor network algorithms have proven to be indispensable for studying some of the most challenging problems in condensed matter physics [14–21] and quantum chemistry [22–27]. More recently they have found new applications in an ever growing list of fields like quantum computation [28–34], machine learning [35–38], numerical analysis and classical differential equation solving [39–47], SAT solving [48,49], and finance [50,51]. Algorithms for tensor networks on linear graph topologies, known as matrix product states (MPS) in the physics literature and tensor trains (TT) in the applied math literature, are extremely well developed and controlled [4,41,52–71], even in the limit of networks with an infinite number of tensors [10,52,53,72–78].

Extensions to tensor network algorithms on general tree topologies, known as tree tensor networks (TTN) in the physics literature and hierarchical Tucker (HT) decompositions in the applied math literature, are similarly well developed and controlled both in the finite [25,79–90] and infinite [91–99] tensor network limits. This can be attributed to a few fortuitous properties of tree tensor networks. Specifically, they are both efficient to contract exactly [100,101] and have simple canonical forms made of locally isometric tensors that define a reduced orthogonal basis at any site in the network [55,56,74,102]. This latter property has many benefits. For instance, it allows for globally optimal truncation with local operations and leads to an elegant formulation of the projector onto the tangent space of the MPS/TT [10,59,66,69,71,103–106] or TTN/HT [83,87–89,107] manifold. Moreover, it makes optimization and evolution algorithms for MPS/TT and TTN/HT very well behaved by allowing one to reduce the global optimization or evolution problem to an alternating series of well-conditioned and stable local updates [1,4]. This is one of the reasons behind the huge success of the celebrated density matrix renormalization group (DMRG) algorithm of White [2,52,53], an extremal eigensolver for MPS/TT that has served as the basis for many subsequent algorithmic advancements.

Unfortunately, for tensor networks that involve loops, the story is not so simple. Some examples of tensor networks with loops are periodic MPS [102,108,109] — also known as tensor chain (TC) or tensor ring (TR) decompositions in the applied math literature [40,110, 111] — and tensor networks with general graph connectivity known as tensor product states (TPS) or projected entangled pair states (PEPS) [112–123]. Contracting tensor networks with loops is generally more costly and can only be done approximately [124], except for in special cases. Moreover, there is no single, obvious gauge that is simple to obtain for performing optimal truncations and well-behaved optimization.

Isometric gauges have been proposed for use on loopy tensor networks that reproduce many of the favorable properties of tensor network gauges on tree topologies [125–130], but their representational power and practical usage is still under investigation and in general they can only be obtained approximately with non-local gauge transformations. Though they do not have all of the benefits of canonical forms on tree tensor networks, a number of local gauge transformations have been proposed for loopy tensor networks with various costs and benefits [116, 120, 131–138]. Local gauge transformations can provide a number of uses in tensor network algorithms: enabling effective truncation of the bond dimension by identifying the most important degrees of freedom [116, 119, 120, 131, 137, 139], improving the conditioning of solvers [121, 132, 133], enhancing the effectiveness of caching environments during optimization [121, 133], and improving analysis of scaling properties of tensor networks [140]. A tensor network gauge that is commonly used for TPS/PEPS has been referred to as the 'super-orthogonal' [131] or 'quasi-canonical' [120, 133] gauge, and is a natural generalization of the 'canonical form' or 'Vidal gauge' originally defined for MPS [55, 56, 74, 75] and TTN [79, 94]. In this work, we will refer to it as the 'Vidal gauge' for both tree tensor networks and more general (loopy) tensor networks. The Vidal gauge for general tensor networks was originally proposed for the purpose of approximately truncating TPS/PEPS in a relatively cheap way, though potentially with a heavy approximation for networks that do not have tree-like correlations. It is the gauge implicitly used in the 'simple update' algorithm [116, 119, 122, 139] for updating or evolving PEPS. Finding better gauges, as well as faster and more reliable methods for gauging TNs, is of utmost importance for improving the conditioning and reliability of tensor network truncation, optimization, and evolution methods — thus expanding their use cases.

Recently, it has been noted that there is a close connection between belief propagation (BP) [141–144], an algorithm originally formulated for performing statistical inference on graphical models, and tensor network contraction [145–152] and gauging [153]. Belief propagation can exactly contract tensor networks on tree graphs (TTN/HT), including path graphs (MPS/TT). Additionally, it is known to work well for tensor networks on graphs that are locally tree-like, such as sparse, random graphs [149]. Here, we make the connection between belief propagation and the commonly used Vidal (or quasi-canonical/super-orthogonal) gauge that was pointed out in Ref. [153] more concrete by proposing an algorithm for using the belief propagation fixed point to gauge general tensor network states into the Vidal gauge. We refer to this new gauging method as belief propagation (BP) gauging. We compare and contrast our new gauging method to existing methods, and argue that it can be viewed as a simplified version of the gauging algorithm introduced in Refs. [131, 133]. We show examples where, for both structured and unstructured networks, our new approach is faster than currently available methods for gauging tensor network states. We also demonstrate its usage for gauging infinite tensor networks and discuss the application of this method for improving the accuracy of evolving tensor networks with 'simple update' [116, 122] gate evolution. Our new BP-based gauging method has the practical advantage that implementations and algorithmic advances of BP (such as the recent [151] or [150, 154]) can be repurposed for gauging tensor networks. We show an example of this in the context of approximately contracting a tensor network operator with a tensor network state. Furthermore, BP is a very simple algorithm based entirely on tensor contractions which is easy to implement efficiently for general tensor networks, something which can be exploited by our new BP-based gauging method.

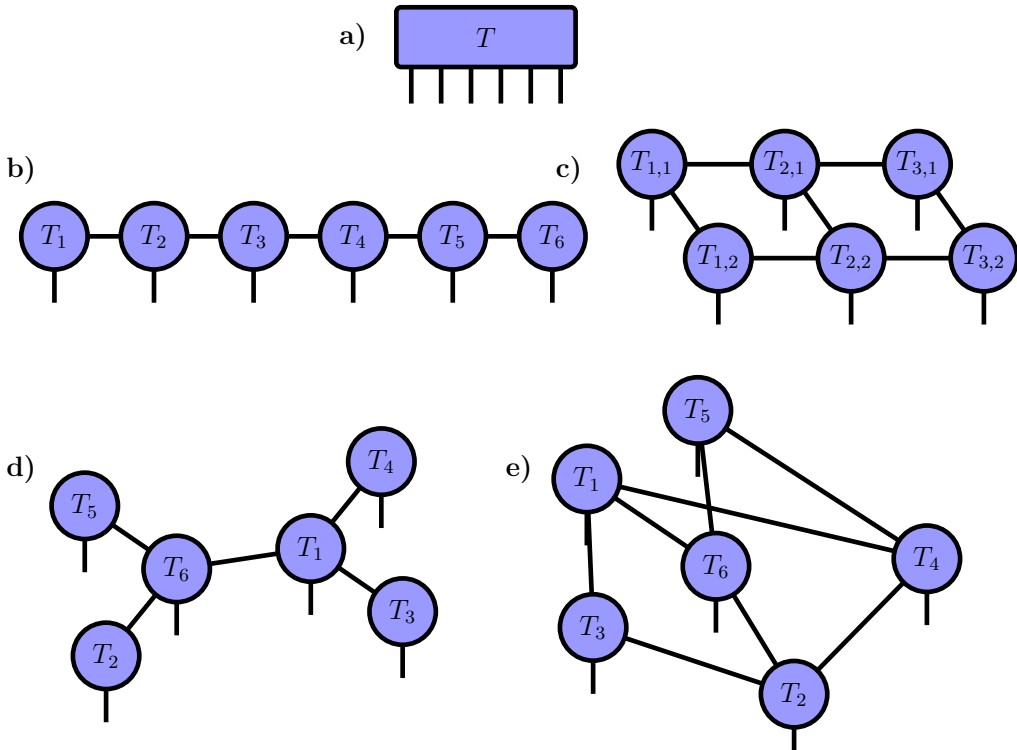

Figure 1: Example tensor network states (TNSs) as decompositions of an order 6 tensor $T$. **a)** Single tensor $T$, **b)** a matrix product state (MPS) or tensor train (TT) decomposition of $T$, **c)** a projected entangled pair state (PEPS) or tensor product state (TPS) decomposition of $T$, **d)** a tree tensor network (TTN) or hierarchical Tucker (HT) decomposition of $T$, and **e)** a generic tensor network state (TNS) decomposition of $T$. Note that here we depict cases where all tensors have one external index, but in general tensors of a TNS can have zero or more than one external index, corresponding to the decomposition of an order $L$ tensor into a number of tensors not equal to $L$.

## 2 The belief propagation gauging method

### 2.1 Tensor network states and the Vidal gauge

We define a tensor network state (TNS) as a connected network of tensors: each vertex $v$ of the network hosts a tensor $T_v$ and the edges $\{e\}$ of the network dictate which tensors share common indices; contraction/summation is implied over those indices. Each tensor of a TNS can also have external indices not common to any other tensors in the network. A TNS with a total of $L$ external indices corresponds to a decomposition of an order $L$ tensor. Fig. 1 illustrates TNS decompositions of an order $L = 6$ tensor into different tensor network structures.

In this work we will demonstrate a new, systematic way to gauge a generic TNS. In general, a gauge transformation of a TNS involves a modification of the site tensors which leaves the overall representation unchanged, i.e. when contracting the full network the result is independent of the transformation. Most commonly, such a transformation involves inserting resolutions of the identity matrix $X_e^{-1} X_e = \mathbb{I}_e$, where $X_e$ is an invertible matrix, on each edge $e$ of the TNS and absorbing the matrices $X_e^{-1}$ and $X_e$ into the incident tensors $T_v$.

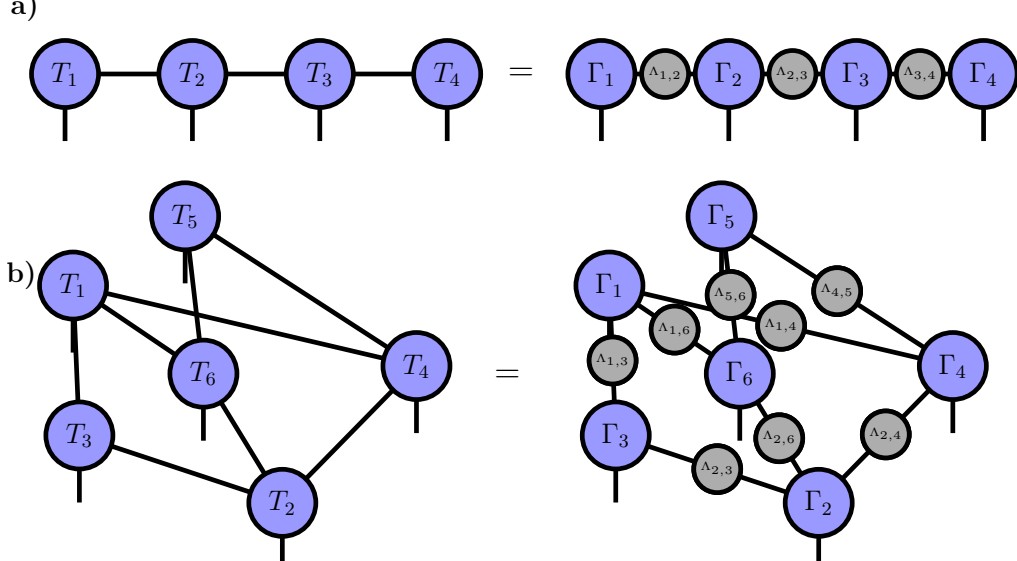

Figure 2: Examples of TNS in the Vidal gauge. **a)** MPS and its equivalent form in the Vidal gauge, and **b)** arbitrary TNS and its equivalent form in the Vidal gauge. Within the Vidal gauge, each tensor $\Gamma_v$ obeys the isometry condition defined by Eqs. (1) and (2).

Our work systematically identifies the gauge transformation necessary to bring a generic TNS into a gauge commonly used in the tensor network literature. In the case of tree topologies, it is commonly referred to as the 'canonical form' or 'Vidal gauge' [55, 56, 74, 75, 79, 94, 96, 97], while for general tensor networks it is the gauge implicitly used in the 'simple update' algorithm [116, 119, 139] and has been called the 'super-orthogonal' [131] or 'quasi-canonical' [120, 121] gauge. In this work, we will refer to it as the 'Vidal gauge' for both tree tensor networks and more general (loopy) tensor networks. This gauge involves a TNS with tensors $\Gamma_v$ on the vertices of the network and non-negative diagonal bond tensors $\Lambda_e$ on the edges of the network. A few examples of TNS in the Vidal gauge are illustrated in Fig. 2. Importantly, within the Vidal gauge, site tensors are 'isometric' upon absorbing all but one of their incident bond tensors. Specifically, by first defining the tensor

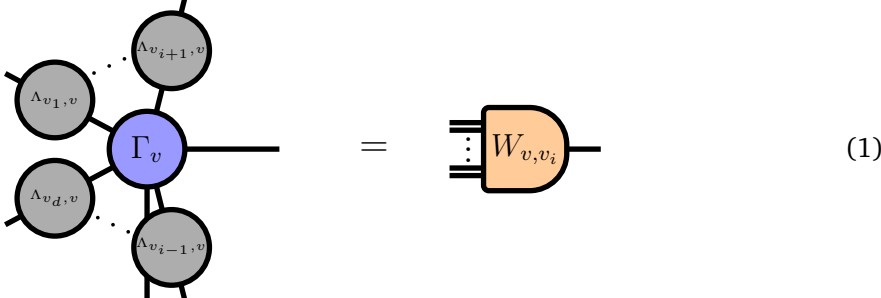
$$\tag{1}$$

this isometric property can be expressed as

$$\tag{2}$$

where $^*$ denotes complex conjugation and the right-hand side of Eq. (2) represents the identity matrix (provided there is an appropriate normalization on $\Gamma_v$ and $\Lambda_e$). The open indices in Eq. (2) are directed along the edge between $v$ and one of its $d$ neighbors $v_i$. The Vidal gauge can

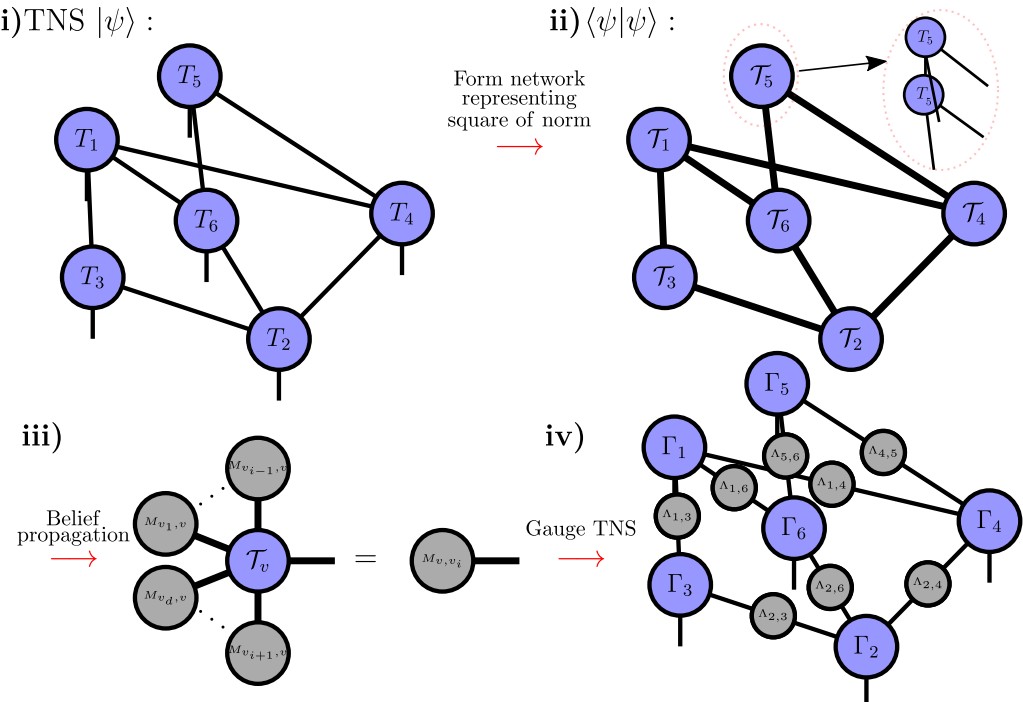

Figure 3: Steps of our belief propagation gauging routine. Starting with a TNS $|\psi\rangle$ the corresponding tensor network representing the square of the norm of the TNS $\langle\psi|\psi\rangle$ is formed. We emphasize that in practice it may be more efficient to keep the tensors in the bra and ket separate and not explicitly contract over the site indices ahead of time (shown by the expanded view of the ringed vertex $\mathcal{T}_5$ of the norm network in **ii)**). Belief propagation is then performed on this network and the fixed point message tensors are used to bring the original TNS into the Vidal gauge. In Section 3.5, we describe a generalization of this procedure to the case where the site tensors $T_v$ of the original network are replaced by sets of tensors. In this case the BP gauging routine is run on a partitioned tensor network comprised by groups of tensors associated with each vertex $v$.

be defined for an arbitrary (finite or infinite) TNS and the bond tensors can be used to provide a rank-one approximation[1] of the environment in a general loopy network — allowing rapid, approximate local updates to be performed on the TNS. When the TNS has a tree structure, the Vidal gauge allows one to define an exact reduced, orthogonal basis at any site in the network. We will introduce a new method for transforming a TNS into the Vidal gauge using belief propagation, independent of the network structure. The essential steps are outlined in Fig. 3 and in the following we will provide the explicit details.

## 2.2 Belief propagation for tensor network states

In order to present our gauging routine we will first introduce the belief propagation (BP) method. BP is a well-established technique for approximating the marginals of the probability

---

[1]Methods like boundary MPS [114, 117, 155] or corner transfer matrix renormalization group (CTMRG) [118, 156–159] exist for obtaining higher-rank approximations of environments in loopy tensor networks, though they have primarily been applied to tensor networks on grids or other regular lattices. Other approximate tensor network contraction methods include a variety of generalizations of BP [143, 147, 150, 152, 154, 160–162], tensor renormalization methods [163–165], and a growing list of other general approximate contraction algorithms [29, 166, 167].

distributions of graphical models [141–144, 154] and has recently been adapted and generalized to tensor networks [145, 146, 149, 150]. Here, we will describe the technique in the context of a 'closed' tensor network, which is a tensor network where there are no external indices and thus its contraction yields a scalar. In our context, the closed network is formed as the square of a TNS — this network is composed of pairs of site tensors $T_v$ and their conjugates $T_v^*$. We will often denote this pair of tensors with the shorthand notation $\mathcal{T}_v$, and refer to this closed network as the 'norm network' of the TNS. For instance, the norm network of a $3 \times 2$ TNS is

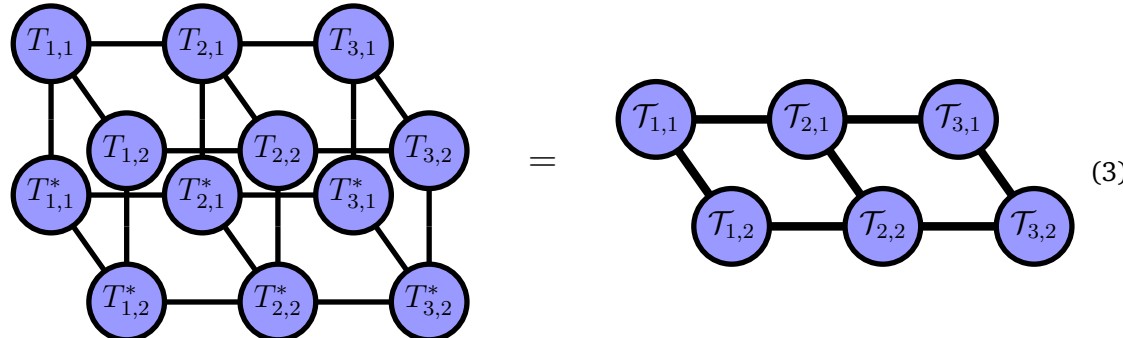

$$\tag{3}$$

where, for efficiency purposes, it is typically advisable to not explicitly contract the two layers together but store the norm network as the structure on the left-hand side of Eq. (3).

We define a set of 'message tensors' of a given norm network $\{M_e, M_{\bar{e}}\}$ for each edge $e$, where $\bar{e}$ denotes the reverse of an edge $e$. We will also use the notation $M_{v,v_i}$, which denotes the message tensor directed along the edge from vertex $v$ to its neighbor $v_i$. The indices of $M_{v,v_i}$ match the indices shared by $\mathcal{T}_v$ and $\mathcal{T}_{v_i}$. In general, the direction of the edge is important and $M_e = M_{v,v_i} \neq M_{\bar{e}} = M_{v_i,v}$. A set of self-consistent equations is defined for the message tensors:

$$M_{v,v_i} = \left( \prod_{j \in \{1,...,i-1,i+1,...,d\}} M_{v_j,v} \right) \mathcal{T}_v, \tag{4}$$

where multiplication of two tensors implies a contraction over any common indices they share. We use the set $N(v) = \{v_1, v_2, ..., v_d\}$ to denote the $d = |N(v)|$ neighbors of $v$, and the product in Eq. (4) runs over all neighbors of $v$ exluding $v_i$. This equation can be expressed diagrammatically as

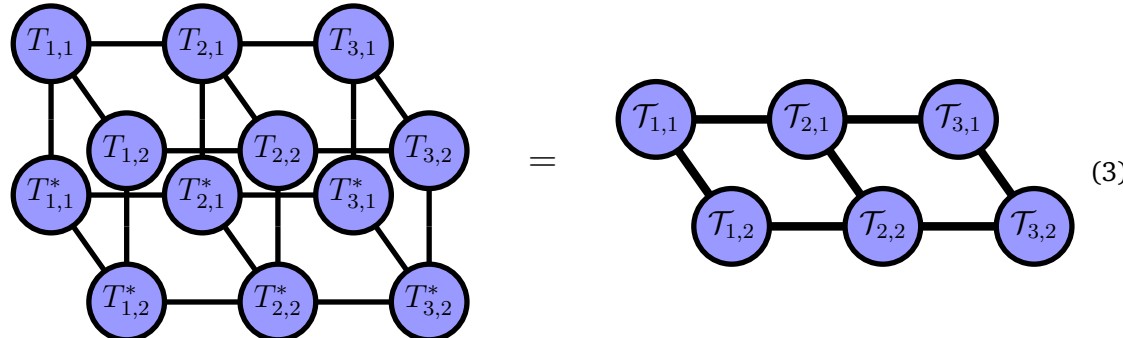

$$\tag{5}$$

Again we emphasize it is generally more efficient to treat $\mathcal{T}_v$ 'lazily' when performing the message tensor update, i.e. as the uncontracted pair $T_v$ and $T_v^*$. Additionally, it is straightforward to generalize to the case where $T_v$ (and $T_v^*$) are replaced by sets of tensors associated with the vertex $v$, which corresponds to a case of BP but with larger groups or partitions of tensors representing $\mathcal{T}_v$. See Section 3.5 for more details and examples of more general TNS structures.

After initializing the message tensors, one can iterate the set of equations defined by Eq. (5) in an attempt to converge them. In Section 3 we explain how convergence can be measured, along with numerical results on the convergence of BP for various example networks.

The converged message tensors then form an approximation for the exact environment for a given vertex $v$. The exact environment would be the result of contracting the full network surrounding $\mathcal{T}_v$: an operation which, for a general network, scales exponentially in the size of the network [48, 124, 168]. Mathematically, the message tensor approximation for the environment can be stated as

$$\prod_{w \neq v} \mathcal{T}_w \approx \lambda_v \prod_{v_i \in N(v)} M_{v_i, v}, \tag{6}$$

where the product on the left-hand side runs over all vertices in the network excluding $v$ and the right-hand side runs over the set $N(v) = \{v_1, v_2, ..., v_d\}$ of $d = |N(v)|$ neighbors of $v$. The scalar $\lambda_v$ is dependent on the normalization of the message tensors and the tensors of the TNS. For illustration, in a 6-site network, an example of Eq. (6) would be

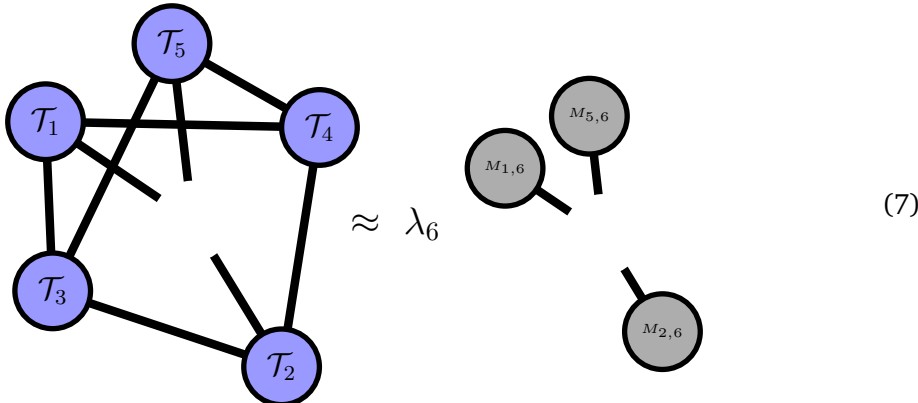

$$\tag{7}$$

This can be used to approximate expectation values (e.g. Eq. 37) or approximately contract a gate with a TNS (e.g. Fig. 5b). Importantly, if the network forms a tree and the message tensors have converged to a fixed point, i.e. they satisfy Eq. (4), then Eq.(6) is exact and no longer an approximation. Next, we will show how belief propagation can be used to transform an arbitrary tensor network state into the Vidal gauge, starting with a review of gauging matrix product states (MPSs) from the perspective of belief propagation.

## 2.3 Using belief propagation to gauge a matrix product state

We will start by reviewing how to transform a matrix product state (MPS) or tensor train (TT) into the canonical form or Vidal gauge [2, 55, 56, 75], but reframed in the language of belief propagation. A MPS is a tensor network with a linear or path graph topology. Labelling the vertices as $v \in \{1, ..., L\}$, each vertex $v$ has neighbors $N(v) = \{v - 1, v + 1\}$ in the bulk of the chain:

$$T_1 - T_2 - T_3 - \cdots - T_L \tag{8}$$

Let us first take a given MPS and combine it with its conjugate to create the closed tensor network which represents the square of its norm (the 'norm network'):

$$\mathcal{T}_1 - \mathcal{T}_2 - \mathcal{T}_3 - \cdots - \mathcal{T}_L = \begin{matrix} T_1 - T_2 - T_3 - \cdots - T_L \\ | \quad | \quad | \quad\quad | \\ T_1^* - T_2^* - T_3^* - \cdots - T_L^* \end{matrix} \tag{9}$$

The self-consistent BP equations are then

$$
\begin{array}{c}
\includegraphics{eq10}
\end{array} \tag{10}
$$

along with corresponding message tensor updates for $\{M_{v,v-1}\}$, where we visualize the equations both in terms of $T_v$ explicitly or grouped into $\mathcal{T}_v$ (though in practice keeping it in terms of $T_v$ is generally more efficient). Importantly, if the initial message tensors used are positive semidefinite then the self-consistency equations are guaranteed to preserve this property at each step.[2] Due to the simple topology of the network, successively applying Eq. (10) from left to right (or right to left for computing $\{M_{v,v-1}\}$) will give convergence of the message tensors after a single update of each message tensor — recovering a standard method for contracting a MPS [2]. Because the network is a tree, these message tensors form an exact representation of the environments of the local norm tensors $\mathcal{T}_v$. After reaching this fixed point, it follows from Eq. (10) that we now have a set of message tensors $\{M_e, M_{\bar{e}}\}$ over the edges $e$ — and their reverses $\bar{e}$ — of the norm network where

$$
\begin{array}{c}
\includegraphics{eq11}
\end{array} \tag{11}
$$

which means that $M_{v-1,v}^{\frac{1}{2}} T_v M_{v,v+1}^{-\frac{1}{2}}$ is an isometric tensor. This corresponds to transforming the MPS into the orthogonal or mixed canonical gauge [2,10]. We have defined the square root of a message tensor $M_e = \left(M_e^{\frac{1}{2}}\right)^{\dagger} M_e^{\frac{1}{2}}$ by performing the eigendecomposition $M_e = U_e^{\dagger} D_e U_e$ and letting $M_e^{\frac{1}{2}} = D_e^{\frac{1}{2}} U_e$, where $U_e$ is unitary and $D_e$ is a positive semidefinite diagonal matrix. Additionally, $M_e^{-\frac{1}{2}} = U_e^{\dagger} D_e^{-\frac{1}{2}}$ where a pseudoinverse may be required if small eigenvalues are present. Note that this definition of the square root is not unique and has a unitary degree of freedom, i.e. we can just as well use $M_e^{\frac{1}{2}} = W_e^{\dagger} D_e^{\frac{1}{2}} U_e$ for any unitary $W_e$.

Now, using the message tensors found from BP we take the following singular value decomposition (SVD)

$$
\begin{array}{c}
\includegraphics{eq12}
\end{array} \tag{12}
$$

which is an approximation if singular values of the bond tensor $\Lambda_{v,v+1}$ are discarded and its dimensions reduced, which amounts to an optimal truncation of the MPS rank or bond dimension according to the 2-norm. The $U$ and $V$ tensors are isometric and satisfy the following properties

$$
\begin{array}{c}
\includegraphics{eq13}
\end{array} \tag{13}
$$

$$
\begin{array}{c}
\includegraphics{eq14}
\end{array} \tag{14}
$$

---

[2]This is proven in Ref. [102] for a MPS, but generalizes to arbitrary networks.

a fact indicated by their semistadium shape. We define the following $\Gamma_v$ tensors as

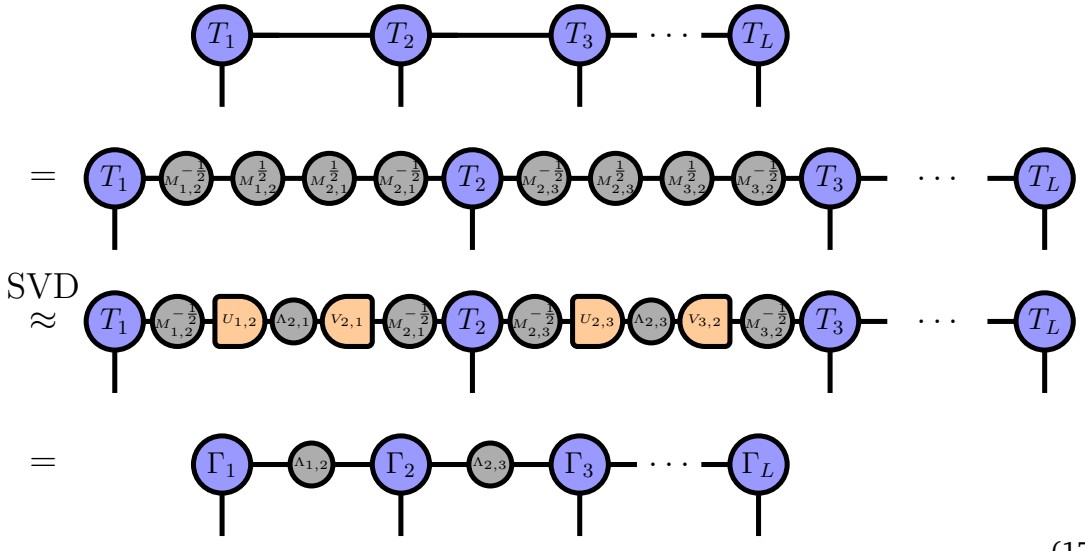

$$(15)$$

and the $\Lambda_{v,v+1}$ bond tensors as

$$(16)$$

These can be directly used to construct a version of the original matrix product state in the Vidal gauge. Specifically, we have transformed our original MPS to a MPS in the Vidal gauge by the following sequence of operations:

$$(17)$$

with equality occurring if no truncation is done during the SVD. Each local tensor in the Vidal gauged MPS now approximately satisfies

$$(18)$$

$$(19)$$

where again equality occurs if no singular values are discarded during the SVD. These identities can be proven by: **i)** substituting the expressions for $\Lambda_{v,v+1}$ and $\Gamma_v$ in Eqs. (15) and (16) into Eqs. (18) and (19), **ii)** utilizing that $U_{v,v+1}^{\dagger}U_{v,v+1}$ and $V_{v,v+1}^{\dagger}V_{v,v+1}$ act as approximate resolutions of the identity up to the singular values truncated in Eq. (12), and **iii)** using the result in Eq. (11) and the isometric constraints Eqs. (13) and (14). The method we have described above can be straightforwardly applied to a tree tensor network state (TTNS), where

it is also possible to find the fixed point message tensors in a single update by performing the message tensor updates in a particular sequence. Once in the Vidal gauge, optimal truncations according to the 2-norm can be performed of the MPS/TTNS to lower the bond dimension (i.e. by truncating according to the singular values as shown in Eq. (12)).

## 2.4 Using belief propagation to gauge a tensor network state

We will now generalize the method for transforming a MPS into the Vidal gauge described in the previous section to demonstrate how to transform a generic tensor network state into the Vidal gauge without any assumptions of a tree-like structure, as illustrated in Fig. 3. Again, one first converges the self-consistent BP equations — see Eq. (5) — over the norm network of the TNS, to obtain the set of message tensors $\{M_e, M_{\bar{e}}\}$. Unlike with tree tensor networks, however, in general there is no direct way to converge the message tensors after a single update and so one must generally rely on iterating the self-consistent BP equations. Upon convergence of these equations, the following identity is true on every edge of the network:

$$ \tag{20} $$

or in other words if the tensors on the top half of the left-hand side are contracted together they form an isometric tensor and the tensors on the bottom half form its conjugate. This relationship can be derived from Eq. (5) by taking the square roots of the message tensors and inverting the square root message tensors on the right-hand side of the equation. Just as for the case of a MPS we define the following:

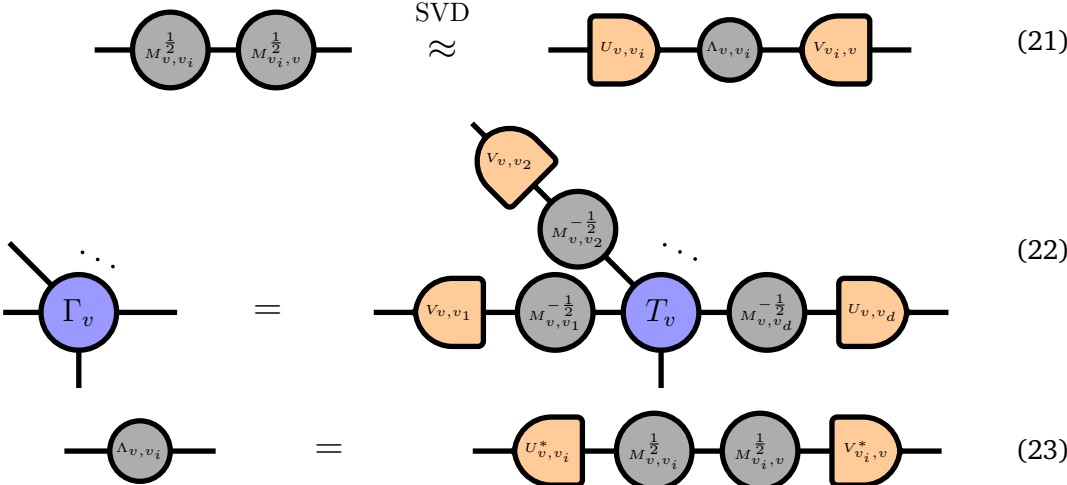

$$ \tag{21} $$

$$ \tag{22} $$

$$ \tag{23} $$

The $U_{v,v_i}$ and $V_{v_i,v}$ matrices again possess the isometric properties described in Eqs. (13) and (14). We can then transform[3] the TNS into the Vidal gauge from the fixed point message tensors and the original TNS using Eqs. (22) and (23). Such a transformation corresponds to: **i)** inserting the resolution of identity $\mathbb{I}_e = M_e^{-\frac{1}{2}} M_e^{\frac{1}{2}} M_{\bar{e}}^{\frac{1}{2}} M_{\bar{e}}^{-\frac{1}{2}}$ along the edges of the original

---

[3]In the case of a tree tensor network state (TTNS) then the TTNS is brought into the canonical form and the $\Lambda_e$ store exactly the singular values of a bipartition of the state across the edge $e$.

TNS, **ii)** performing the SVD in Eq. (21), and **iii)** absorbing the inverse square root message tensors $M_e^{-\frac{1}{2}}$ along with the $U_e$ and $V_e$ tensors obtained from the SVD onto the on-site tensors $T_v$, analogous to the steps shown in Eq. (17) for gauging a MPS. For the example of a 6-site TNS the result is

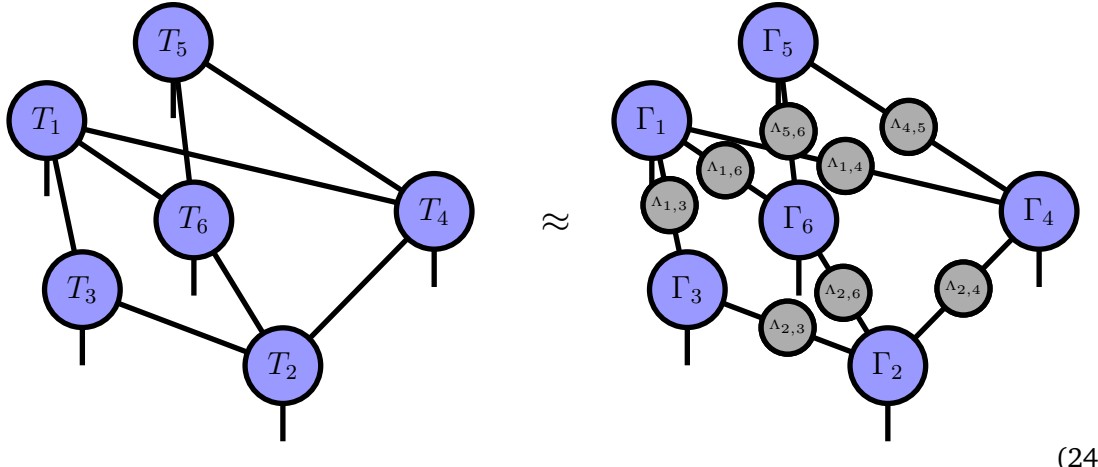

$$(24)$$

which, if no truncation is done during the SVD, becomes an equality. Note that SVD truncations are not optimal for loopy networks, and are only valid up to the BP approximation (i.e. can be close to optimal for TNS with tree-like correlations).

Assuming the message tensors used satisfy the fixed point criterion in Eq. (5) and no truncation is done then the right-hand side of Eq. (24) is provably in the Vidal gauge: Eqs. (1) and (2) are valid along all edges of the network. The proof of this is essentially analogous to that for a MPS and can be achieved by **i)** substitution of the expressions for $\Gamma_v$ and $\Lambda_e$ into the tensor network in the Vidal gauge (such as the right-hand side of Eq. (24)), **ii)** utilizing that $U_e^\dagger U_e$ and $V_e^\dagger V_e$ act as approximate resolutions of the identity up to the singular values truncated in Eq. (21), and **iii)** invoking the identity in Eq. (20) and the isometric constraints Eqs. (13) and (14).

Eqs. (21), (22), and (23) define an algorithm for bringing an arbitrary TNS into the Vidal gauge. This can be summarized by the following steps:

---

**The belief propagation (BP) gauging routine**

- Start with a TNS consisting of site tensors $\{T_v\}$. Form the closed network corresponding to the square of the norm of the TNS.

- Initialize the belief propagation message tensors $\{M_e, M_{\bar{e}}\}$ of the normed network to arbitrary, positive definite matrices.[4]

- Perform iterations of belief propagation, where each iteration corresponds to an update of each message tensor via Eq. (5), up to some stopping criteria.

- Gauge the TNS with the resulting message tensors via Eqs. (21), (22), and (23). This yields the TNS in the Vidal gauge with site tensors $\{\Gamma_v\}$ and bond tensors $\{\Lambda_e\}$.

---

The less truncation performed during the SVD in Eq. (21) and the more iterations of belief propagation performed, the closer the tensors in the TNS will be to satisfying the isometric

---

[4]While initializing to positive definite matrices may not be strictly necessary it guarantees the message tensors are positive semidefinite at each step and therefore their square root can be taken.

conditions in Eqs. (1) and (2). We refer to our algorithm as belief propagation gauging or BP gauging for short.

Note the very close similarity to the steps of BP gauging, such as the definitions of $\Gamma_v$ and $\Lambda_e$ in Eqs. (21), (22), and (23), to the steps of the gauging method introduced in Refs. [131, 133], which also transforms a general tensor network state into the Vidal gauge and which we refer to as 'eager gauging'. A key distinction from that method is that in BP gauging, the gauge transformation is only performed *once* at the end, after belief propagation is performed on the original network to a desired level of convergence of the message tensors. In contrast, gauge transformations are performed at every iteration in eager gauging. In fact, in the language of belief propagation, the eager gauging method can be interpreted as alternating steps of running an iteration of belief propagation on the symmetric gauge of the network (defined in Section 2.6 below) and then performing the gauge transformation defined in Eqs. (21), (22), and (23) on the updated message tensors. It turns out that for the sake of gauging a tensor network into the Vidal gauge, performing gauge transformations after each step of BP besides the final step is extraneous and can be avoided altogether. We summarize the eager gauging algorithm defined in Refs. [131, 133] in the language of BP in Appendix B. Numerical results in Sec. 3.2 corroborate that the gauging methods have the same convergence properties but that BP gauging is faster because it requires fewer operations at each iteration.

## 2.5 Using square root belief propagation to gauge a tensor network state

An alternative standard procedure for gauging a MPS or TTN into the orthogonal or canonical form involves taking QR decompositions (or some other orthogonal decomposition) of what would be, in the language of belief propagation, the square roots of the message tensors absorbed into the site tensors [2]. This amounts to performing the MPS message tensor update shown in Eq. (10) using a QR decomposition $M_{v-1,v}^{\frac{1}{2}} T_v = Q_{v,v+1} M_{v,v+1}^{\frac{1}{2}}$ (or $T_v M_{v+1,v}^{\frac{1}{2}} = M_{v-1,v}^{\frac{1}{2}} Q_{v,v-1}$ in the case of a right to left update). After squaring, the resulting message tensors would still satisfy the relations in Eqs. (10) and (11). This has the advantage that the square root message tensors can be found to higher precision, though at the cost of performing QR decompositions. An analogous 'square root' belief propagation update can be performed for arbitrary tensor networks[5] using QR decompositions to perform square root message tensor updates:

$$
\cdots \quad \cdots \\
\underset{M_{v_1,v}^{\frac{1}{2}} \\ M_{v_d,v}^{\frac{1}{2}}}{\cdots} T_v = Q_{v,v_i} M_{v,v_i}^{\frac{1}{2}} \tag{25}
$$

which can be iterated to find a set of fixed point square root message tensors $\left\{ M_e^{\frac{1}{2}}, M_{\tilde{e}}^{\frac{1}{2}} \right\}$. These can then be directly used for transforming the network into the Vidal gauge with Eqs. (21), (22), and (23). As in the MPS case this can have the advantage of providing higher numerical precision. To avoid some of the higher cost due to performing QR decompositions (or some other orthogonal decomposition), one can first converge using the standard message tensor updates from Eq. (5) and then switch to the square root message tensor update in Eq. (25), similar to procedures formulated for gauging MPS [7, 10, 159]. Note that this is closely related to the 'simple update gauging' method which we review in Appendices A and B. Specifically, a square root BP update step is analogous to steps **i-iii)** of Fig. 11 but, unlike in simple update gauging, regauging is only performed in the final iteration after the square root message tensors are converged. This avoids repeatedly performing gauge transformations at

---

[5]The authors would like to thank Johnnie Gray for pointing this out to us during the preparation of this work.

each message tensor update, which is done at every bond update in simple update gauging (steps **iv-viii)** of Fig. 11).

## 2.6 The symmetric gauge

Here we will introduce a useful gauge for a TNS that is very closely related to the Vidal gauge. Specifically, given a TNS in the Vidal gauge, it is straightforward to define the square roots of the bond tensors:

$$\tag{26}$$

and absorb them onto the site tensors $\Gamma_v$:

$$\tag{27}$$

using the example of a degree $d = 4$ tensor above. We can then recover a TNS without bond tensors:

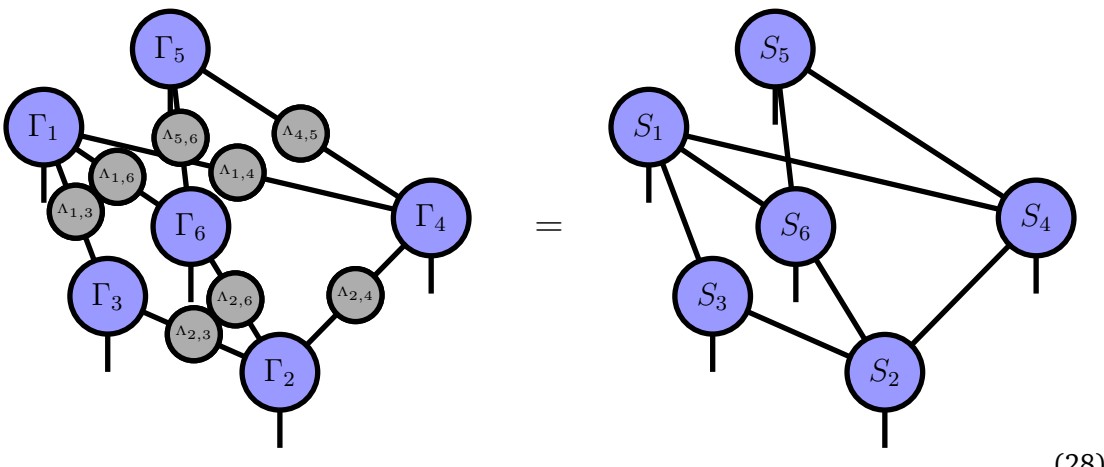

$$\tag{28}$$

This gauge has been introduced previously in the tensor network literature [116, 133, 153]. In the following, we will refer to this as the 'symmetric gauge'. An important aspect of this form is that if we have a state in the Vidal gauge, then transforming to the symmetric gauge and running belief propagation will yield the fixed point message tensors $\{M_e = M_{\bar{e}} = \Lambda_e\}$. In other words, for a TNS in the symmetric gauge, the message tensors are the same in both directions along a given edge $e$, and equivalent to the positive diagonal bond tensors $\{\Lambda_e\}$ of the Vidal gauge of the same TNS. This property of the symmetric gauge was previously pointed out in Ref. [153]. The proof of this can be done by writing down the self-consistency equation for the message tensors of the norm network in the symmetric gauge (Eq. (5)), substituting in the definition of the symmetric gauge tensors (Eq. (27)), and observing that it is equivalent to the Vidal gauge isometric condition defined in Eqs. (1) and (2) if $M_e = M_{\bar{e}} = \Lambda_e$.

# 3 Applications and benchmarks of belief propagation gauging

## 3.1 Quantifying the distance to the Vidal gauge of a tensor network state

In order to detail our numerical results it will be important to quantify how closely a state in the Vidal gauge actually obeys the isometry condition in Eqs. (1) and (2). This is directly dependent on how close the message tensors used to gauge the original state are to their fixed point and, in a general numerical implementation, they will only be approximately at the fixed point and thus the isometry condition in Eqs. (1) and (2) will only be obeyed approximately.

To treat this we introduce a new quantifier $\mathcal{C}$ which we refer to as the 'distance to the Vidal gauge'. We will use this to measure the effectiveness of various algorithms, including ours, at bringing a TNS into the Vidal gauge. Given a TNS in the Vidal gauge — i.e. consisting of bond tensors $\{\Lambda_e\}$ and site tensors $\{\Gamma_v\}$ — $\mathcal{C}$ is defined as

$$\mathcal{C} \;=\; \frac{1}{2|E|}\sum_v \; \sum_{i=1}^{|N(v)|} \left\|\; \cdots \; - \; \square \;\right\|_{1'} \tag{29}$$

where we have introduced the squares of the bond tensors

$$-\!\!\bigcirc\!\!\!\Lambda^2_{v,v_i}\!\!\!-\!\!\! \;=\; -\!\!\bigcirc\!\!\!\Lambda_{v,v_i}\!\!\!-\!\!\bigcirc\!\!\!\Lambda_{v,v_i}\!\!\!- \tag{30}$$

Equation (29) essentially measures the average degree to which Eqs. (1) and (2) are satisfied by the tensors in the TNS. Here, $\|A - B\|_{1'} = \left\|\frac{A}{\mathrm{Tr}(A)} - \frac{B}{\mathrm{Tr}(B)}\right\|_1$ and $\|M\|_1$ represents the trace norm of a matrix $M$. Both matrices in Eq. (29) are scaled to trace unity in order to remove any dependence of $\mathcal{C}$ on the normalization of the network.

Importantly, one can accurately determine the order of magnitude of $\mathcal{C}$ — which we denote with $O(\mathcal{C})$ — by assessing the distance of the message tensors found via belief propagation from their fixed point. This allows one to target a precscribed $O(\mathcal{C})$ in belief propagation gauging while avoiding having to explicitly gauge the state and compute the isometries to measure $\mathcal{C}$ at every step of belief propagation. Specifically,

$$O\!\left(\mathcal{C}^{(n)}\right) \approx \frac{1}{2|E|}\sum_v \sum_{i=1}^{|N(v)|} \left\|M^{(n-1)}_{v,v_i} - M^{(n)}_{v,v_i}\right\|_{1'}, \tag{31}$$

where $n$ is the current iteration of BP. This can be derived using Eqs. (5), (21), (22), and (23) and comparing to Eq. (29).[6] This allow us to use BP to gauge a TNS to a prescribed order of magnitude of $\mathcal{C}$ without any significant computational overhead from checking the value of $\mathcal{C}$.

## 3.2 Accelerating tensor network gauging with belief propagation

Here we perform numerical simulations using the ITensorNetworks.jl package [169], demonstrating the efficacy of the BP gauging method for gauging general tensor network states. For

---

[6]The following reasonable assumptions are necessary in this derivation: $O(\mathcal{C}^{(n)}) \approx O(\mathcal{C}^{(n+1)})$ and $O\!\left(\left\|\frac{A}{\mathrm{Tr}(A)} - \frac{B}{\mathrm{Tr}(B)}\right\|_1\right) \approx O\!\left(\left\|\frac{B^{-1/2}AB^{-1/2}}{\mathrm{Tr}(B^{-1/2}AB^{-1/2})} - \mathbb{I}\right\|_1\right).$

a given TNS we will repeatedly run iterations of belief propagation on the corresponding norm network. We expect to observe the message tensors found after an increasing number of iterations will bring the TNS closer and closer to the Vidal gauge, i.e. they can be used to transform the TNS to a state with an increasingly small value of $\mathcal{C}$ defined in Section 3.1. We will benchmark the convergence of $\mathcal{C}$ with belief propagation iterations against existing algorithms for bringing a TNS into the Vidal gauge.

The first of these algorithms we term *simple update gauging* [116, 120, 153]. This routine starts with a TNS in the Vidal gauge and lowers the corresponding value of $\mathcal{C}$ by repeatedly performing a simple update with 'identity' gates on the edges of the network — the simple update procedure is pictured in Fig. 5, with more extensive details in Appendices A and B. The second algorithm, which we term *eager gauging* [131, 133], is very closely related to our algorithm, as we discussed at the end of Section 2.4. Instead of an iteration of BP, each iteration of eager gauging involves alternating iterations of BP interspersed with gauge transformations. In Appendix B we provide explicit details of the steps involved in both the eager and simple update gauging routines.

The runtime and convergence of these algorithms is dependent on the order in which the message tensors are updated in a given iteration, which is called the schedule. This schedule can either be 'synchronous' or 'asynchronous', also referred to as 'serial' or 'sequential' schedules. In the case of synchronous schedules, all message tensors (or site and bond tensors in the case of simple update or eager gauging) are updated at once given their current state. For asynchronous or serial schedules, message tensors (or site and bond tensors) are updated one at a time, so that message tensor updates can make use of previously updated message tensors within an iteration. Naturally, serial scheduling implies that the order of edges to perform updates over is important while for synchronous scheduling it is not. When a serial schedule is chosen well, it can require fewer message tensor updates to converge [170, 171], though synchronous schedules allow for performing message tenors updates simultaneously in parallel (something which we do not take advantage of in our current benchmarks) . In order to properly compare the various tensor network gauging algorithms, we use a consistent schedule across the different algorithms. In particular, we use a hybrid synchronous-serial schedule where we split the edges of the TNS into a minimum number of groups where edges in the same group don't share common vertices. This corresponds to the grouping of gates/edges in a Suzuki-Trotter decomposition [172] of a nearest-neighbor Hamiltonian on the graph of the tensor network.[7] In a given iteration, we then perform serial updates between edge groups but synchronous updates within the same group. In the case of BP or eager gauging, when a given edge of the TNS appears in the update schedule, we perform a synchronous update of the forward and backward message tensors via Eq. (4).

Fig. 4 displays our benchmark of these gauging routines for a variety of different tensor network states. Specifically, in Fig. 4a-b we consider a square lattice TNS composed of tensors with random elements that are normally distributed around zero. We show how the time to reach $\mathcal{C} \leq 10^{-10}$ scales with system size and bond dimension. We can see that BP gauging is faster than both simple update gauging and eager gauging for all bond dimensions and system sizes. As expected, all of the methods have a dominant scaling of $\mathcal{O}(N\chi^{z+1})$, where $\chi$ is the bond dimension, $z$ is the largest coordination number in the network, and $N$ is the number of sites in the network with coordination number $z$ (where $N = L^2$ in Fig. 4a-b for an $L \times L$ square lattice).

We can explain the performance improvement of BP gauging over the other methods based on counting the number of operations that scale as $O(\chi^{z+1})$ in each iteration of the different

---

[7]A fully synchronous sequence prevents simple update gauging from converging as synchronous updates of the state are not compatible along edges which share common vertices. Hence, we adopt this hybrid sequence to compare the algorithms.

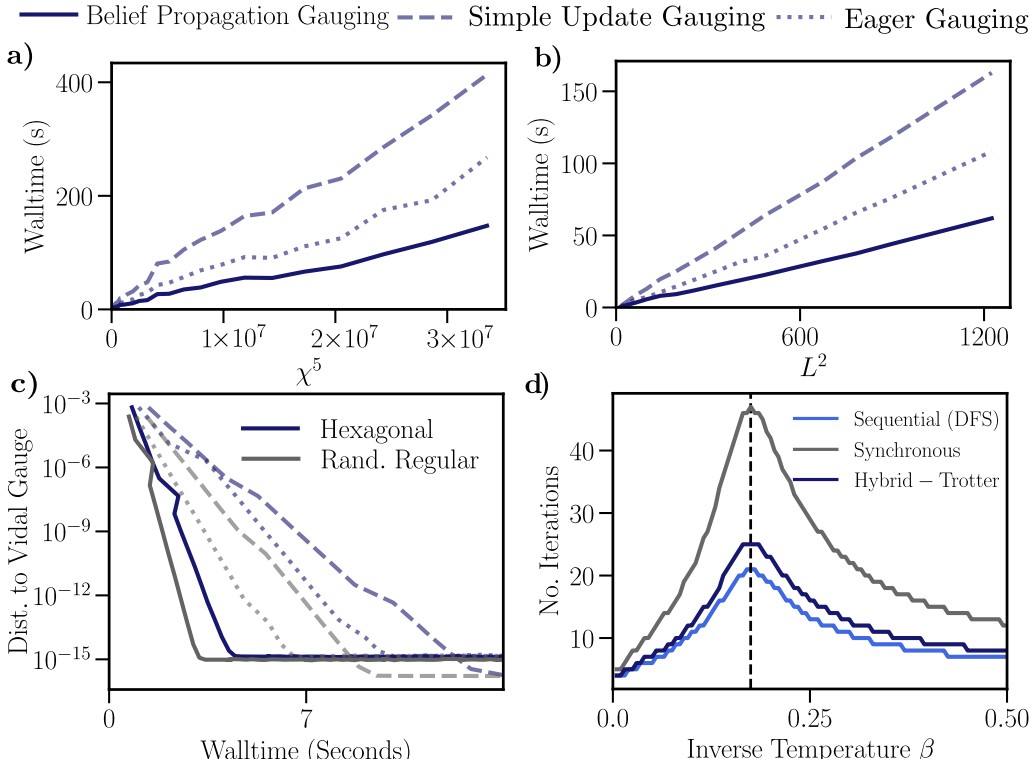

Figure 4: **a-b)** Time to reach $\mathcal{C} = 10^{-10}$ for a random TNS — with tensor elements drawn from the normal distribution of mean 0 and standard deviation 1 — of bond dimension $\chi$ on an $L \times L$ square lattice. **a)** Scaling with $\chi^5$ for $L = 12$. The plot ranges to $\chi = 32$ where we see that eager gauging and simple update gauging take 1.82 and 2.81 times longer than belief propagation gauging respectively. **b)** Scaling with $L^2$ for $\chi = 15$. The plot ranges to $L = 35$ where we see that eager gauging and simple update gauging take 1.75 and 2.63 times longer than belief propagation gauging respectively. **c)** Distance to the Vidal gauge $\mathcal{C}$ versus walltime for gauging a random tensor network state with elements drawn from the normal distribution of mean 0 and standard deviation 1. The bond dimension of the TNS is $\chi = 20$ and two lattices are shown: the hexagonal TNS for 10 rows and 10 columns of hexagons, and a 200 site random regular graph of fixed degree 3. In plots **a-c)** different line styles correspond to different gauging methods (with the legend at the top of the figure) and all walltimes are in seconds. Iterations of each algorithm are performed using a fixed Suzuki-Trotter decomposition-inspired schedule of updates over the edges of the TNS (see main text for more details). **d)** Number of iterations required to reach $\mathcal{C} = 10^{-8}$ for the TNS corresponding to the square root of the classical Ising partition function (which has bond dimension 2) versus inverse temperature $\beta$. A longitudinal field of $h = 0.5$ is included. Results are for a $20 \times 20 \times 20$ cubic lattice. Different colors correspond to different update schedules for the message tensors in a given iteration - with further details in the main text - and the total walltime is directly proportional to the number of iterations.

algorithms. We should emphasize, for a fixed update schedule, the number of iterations required by each algorithm to reach a given $\mathcal{C}$ is identical as each step of the algorithm updates the state in an analogous way, and all of them can be viewed as variations of belief propagation. Thus the performance is entirely based on the timing of a single iteration of each algorithm, up to the fact that the final step of BP gauging will involve an additional step of gauging using

the fixed point message tensors. Each iteration is dominated by a number of operations that scale with the bond dimension $\chi$ as $O(\chi^{z+1})$ in the limit of large $\chi$. The extraneous gauge transformations at each iteration of eager gauging means that there are approximately $z$ extra tensor contractions that are of $O(\chi^{z+1})$ which are not performed by BP gauging.

In simple update, in the limit of large $\chi$ the cost of each iteration is dominated by tensor contractions that scale as $O(\chi^{z+1})$ as well as QR decompositions that scale as $O(\chi^{z+1})$. Like eager gauging, simple update gauging has on the order of $z$ extra tensor contractions that scale as $O(\chi^{z+1})$ compared to BP gauging. Additionally, a QR decomposition of $O(\chi^{z+1})$ generally has a higher constant prefactor than a tensor contraction of $O(\chi^{z+1})$, which also contributes to the higher runtime per iteration compared to BP gauging.

In Fig. 4c we consider random TNSs (again, with normally distributed elements) of different geometries: a hexagonal lattice grid consisting of $10 \times 10$ rows and columns of hexagons and a random regular graph of 200 sites and degree $z = 3$. The distance to the Vidal gauge decreases exponentially with walltime in each algorithm, with BP gauging displaying the largest rate of decay (due to the faster speed of each iteration as argued above).

Finally, in Fig. 4d, we construct the TNS corresponding to the square root of the partition function of the classical Ising model with a longitudinal field on a $20 \times 20 \times 20$ cubic lattice (see Refs. [149, 173] for explicit details on how to construct this state for an arbitrary network). We focus solely on BP gauging and consider the effect of different update schedules for the convergence of the TNS, which has bond dimension $\chi = 2$, to the Vidal gauge. All schedules take the longest to converge at the critical point due to the long-range correlations present in the state. We compare three different schedules: i) fully synchronous updates, ii) the hybrid synchronous-serial schedule inspired by the Suzuki-Trotter decomposition described above, and iii) serial updates based on our custom sequence. We note that the worst performance is observed for fully synchronous update schedules which is consistent with that seen in the BP literature [170, 171]. Here, we find our custom schedule — which is closely related to schedules proposed in [174–176] — is best for the performance of BP gauging. It is based on finding a forest cover of the network [177], performing depth first search (DFS) to find an update sequence for each of the trees in the forests, and concatenating the resulting sequences to get a full sequence for the network. We use a breadth first search to construct the spanning trees which make up the forests which typically finds comb-like trees on grid graphs. This is similar to the spanning trees considered in [174–176]. For a tree tensor network there is only one forest in the forest cover and one tree within that forest. Thus the schedule reproduces the known result that running BP and finding the Vidal gauge only requires a single iteration of BP. For a loopy network, an iteration of BP involves iterating through the trees of the forest cover. We should also point out a wide range of different serial schedules have also been proposed in the BP literature with various advantages and disadvantages [171, 174–176] — comparing these to each other and our custom schedule in the context of tensor network gauging is a topic of future research.

## 3.3 Approximate gate application with belief propagation

A common method for performing gate evolution of a TNS is the simple update (SU) method [116, 123, 178–180], which we summarize in Fig. 5a and Appendix A. The SU gate application method implicitly uses an approximate Vidal (or quasi-canonical/super-orthogonal) gauge. This is the routine at the heart of simple update gauging (which is equivalent to performing SU with identity gates, see Fig. 11 in Appendix A). It is based on an approximation of the environment as a product of matrices, which is exact on trees [79, 94] and can be a good approximation for systems with tree-like correlations. For systems with non-tree-like correlations, such as strongly correlated systems on regular lattices with small loop structures (like square lattices), it is commonly used to provide starting states and rank-one reference points for more

demanding but accurate TNS calculations. Examples of more controlled and accurate methods for evolving or optimizing TNS are 'full update' gate evolution [117, 119, 121, 132, 181] and variational optimization or gradient descent [114, 182–185] based on higher-rank approximations of the environment (which are commonly approximated as MPS).

Importantly, the simple update procedure can actually be performed on a tensor network state in an *arbitrary* gauge by using the message tensors found from belief propagation as the rank-one approximation of the environment. In Fig. 5b we summarize this procedure. This is equivalent to performing a 'full update' [117, 119, 121, 132, 181] with the message tensors as the environment, but the rank-one nature of the environment allows one to solve the fidelity optimization directly instead of iteratively, which is much faster in practice. We emphasize that, if the BP message tensors and Vidal gauge bond tensors are similarly accurate, simple update with BP message tensors is equivalent to performing SU in the Vidal gauge, which is summarized in Fig. 5a. In the Vidal gauge, the bond tensors act as the rank-one approximations of the environment, analogous to the role of message tensors (and in fact are just message tensors in a different gauge, which is at the heart of the BP gauging routine introduced in Section 2.4). Note that both of these routines can benefit from efficiency improvements by performing a 'reduced tensor' SVD (see Appendix A for explicit examples in the context of SU in the Vidal gauge, with straightforward translations of the same procedures to performing SU in arbitrary gauges with BP message tensors).

Finally, we would like to point out that repeated applications of the BP simuple update procedure over the bonds of a TNS with identity gates will transform the TNS to the symmetric gauge (defined in Section 2.6). This is analogous to simple update gauging, where repeated application of identity gates with simple update via Fig. 5a will bring a TNS into the Vidal gauge.

## 3.4 Improving the accuracy of tensor network evolution with regauging

In this section we describe an application of our new BP gauging method to improving the accuracy of TNS gate evolution with the simple update method. Other gauging methods like simple update gauging [116, 120] and eager gauging [122, 131, 133], which we have shown reach the same fixed point as BP gauging, can perform the same task (though they are potentially slower to reach the fixed point compared to BP gauging, as evidenced by our benchmarks in Section 3.2).

As demonstrated by the simple update gauging routine, if enough identity gates are applied using the simple update routine the state will converge to the Vidal gauge, where it approximately satisfies the orthogonality condition defined in Eqs. (1) and (2). For time evolution of a TNS with a Trotter circuit, in the limit of small Trotter steps and truncations, the gates are approximately identity and the gate applications act to both perform the evolution and improve the Vidal gauge orthogonality conditions of the state. However, if gates far from the identity are applied (such as in a generic quantum circuit or for large Trotter step sizes) or if significant truncation is performed during the gate evolution, the state may stray from satisfying the orthogonality conditions. This can lead to a loss in accuracy during the gate evolution because the environments aren't the optimal rank-one environments.

To remedy this situation, one can try to find improved rank-one environments for performing the simple update gate evolution. One strategy would be to run the belief propagation (BP) tensor network contraction algorithm on the norm network of the TNS in the same spirit as Ref. [150], which focused on using BP to compute rank-one environments for calculating expectation values and performing energy optimization of tensor networks on sparse graphs. For this algorithm, one would run BP on the norm network of the state being evolved and use the fixed point message tensors as rank-one approximations of the environment in order to perform gate evolution with the 'full update' algorithm or BP-variant of simple update (which

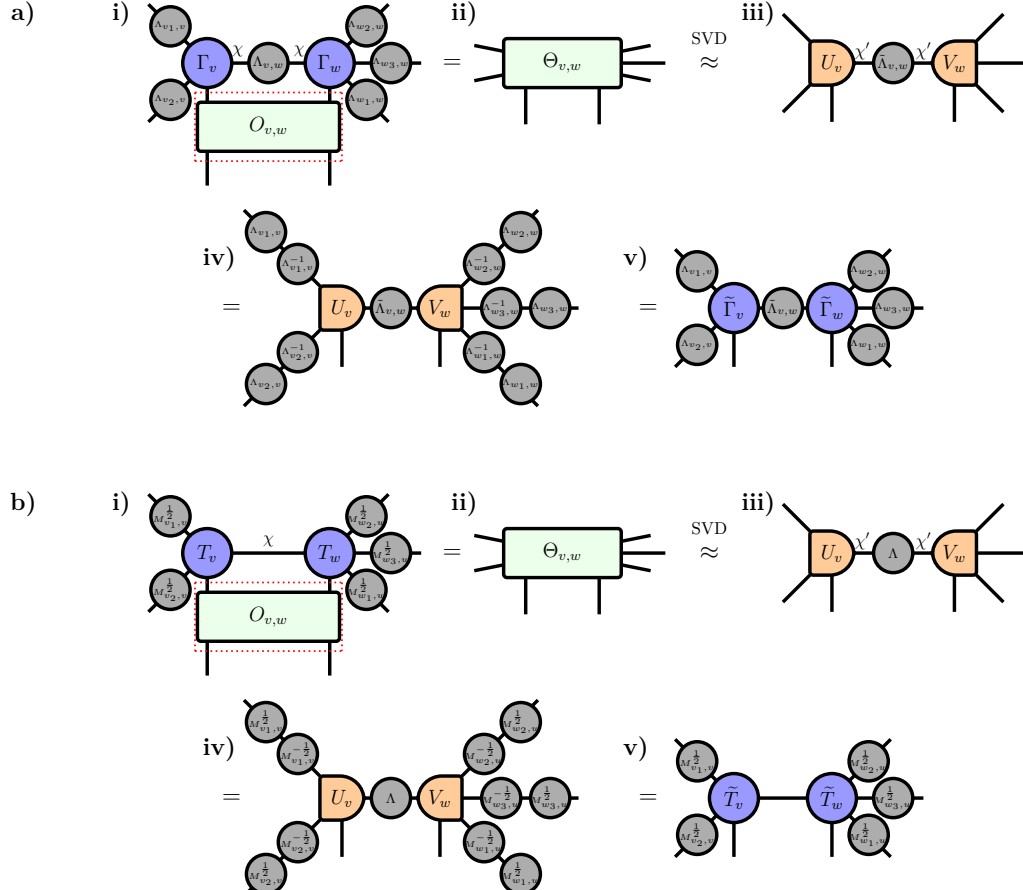

Figure 5: **a)** Simple update gate evolution in the Vidal gauge. **i) - ii)** The bond tensors and gate $O_{v,w}$ (marked by the red dotted line) are combined into a single composite tensor $\Theta_{v,w}$. **ii) - iii)** $\Theta_{v,w}$ is SVDd and singular values of the resulting bond tensor $\tilde{\Lambda}_{v,w}$ can be discarded in order to truncate the bond down to a desired dimension $\chi'$. **iii) - iv)** Resolutions of identity, using the original bond tensors, are inserted on the exposed edges. **iv) - v)** The inverse bond tensors are absorbed, resulting in the updated site tensors $\tilde{\Gamma}_v$ and $\tilde{\Gamma}_w$ and the bond tensor $\tilde{\Lambda}_{v,w}$. **b)** The analog of simple update gate evolution using BP message tensors, equivalent to 'full update' gate evolution where the environment is products of message tensors. The steps are analogous to the simple update procedure for the Vidal gauge. The new message tensors on the bond $v, w$ is taken to be $\tilde{M}_{v,w} = \tilde{M}_{w,v} = \Lambda$, the diagonal tensor from the SVD whose square root is also absorbed by $U_v$ and $V_w$. **a)** and **b)** are equivalent updates of the state and represent ideal rank-one gate applications if, before applying the gates, either the state in **a)** is regauged using one of the Vidal gauging methods or BP is run on the state in **b)** to get updated fixed point message tensors. More efficient versions of these procedures (with and without the gate) are pictured in Figs. 10 and 11.

are entirely equivalent in the case of BP) that we show in Fig. 5b.

Based on the equivalence between the BP fixed point and the Vidal gauge that was first pointed out in Ref. [153] and further solidified by our new BP gauging algorithm, however, this is equivalent to regauging the tensor network into the Vidal gauge and then performing the simple update gate application with the regauged $\Gamma_v$ and $\Lambda_e$ tensors. This 'regauging' technique was (as far as we know) first proposed in [122]. In that work, the authors proposed to use

what we refer to in this work as 'eager gauging' to regauge the network into the Vidal gauge to improve the accuracy of simple update gate evolution, but any gauging method (simple update gauging, eager gauging, or our new belief propagation gauging) can be used. This regauging can be performed at every gate application, in which case it is equivalent to performing the entire evolution with BP message tensors as the environments (either via full update or the version in Fig. 5b). Alternatively, it can be performed only when it is detected that the Vidal gauge orthogonality constraints are no longer accurate up to a certain threshold and therefore will affect the accuracy of the gate application.

We run several dynamical simulations where we apply two-site gates to a TNS in the Vidal gauge with simple update (Fig. 5a) and periodically regauge with BP gauging. This is equivalent to applying two-site gates to a TNS in an arbitrary gauge using message tensor environments (Fig. 5b), while periodically re-running belief propagation to improve the message tensors. We consider two scenarios: **a)** imaginary time evolution of the TNS towards the ground state of the 2D transverse field Ising model at criticality and **b)** repeatedly applying layers of random nearest-neighbour two-site unitary gates to the TNS. We choose 2D square lattices solely because we can use boundary MPS [114, 155] to accurately compute the energy and fidelities for moderately large lattice sizes.

For imaginary time evolution, the Hamiltonian on an $L \times L$ open boundary lattice reads

$$H = \sum_{\langle v,v' \rangle} \sigma_v^x \sigma_{v'}^x - g \sum_v \sigma_v^z = \sum_{\langle v,v' \rangle} h_{v,v'} + \sum_v h_{v'}, \tag{32}$$

where the first summation runs over the nearest neighbour pairs in the lattice and the second runs over all the sites of the lattices. The propagator is Trotterized into a series of two-site gates as

$$U(\Delta\beta) = \exp(-\Delta\beta H)$$
$$= \left( \prod_{\langle v,v' \rangle} \exp\left(-\frac{\Delta\beta}{2} h_{v,v'}\right) \right) \left( \prod_v \exp(-\Delta\beta h_v) \right) \left( \prod_{\langle v,v' \rangle} \exp\left(-\frac{\Delta\beta}{2} h_{v,v'}\right) \right) + \mathcal{O}(\Delta\beta^2). \tag{33}$$

We apply the two-site gates using the simple update procedure as described in Appendix A, truncating to a specified maximum bond dimension during the SVD. We perform belief propagation gauging after the application of a certain number of gates. The case when no gauging is performed is a common method of doing imaginary time evolution in the literature [116, 153, 180, 186] and we compare to that here.

In Fig. 6a we plot the energy of the state obtained under a sequence of applications of $U(\Delta\beta)$ for $g = 3.0$, which is close to the critical point of the 2D transverse field Ising model. We start from the 'Néel' product state, where neighboring spins are polarized in opposite directions along the spin z-axis. Targeting $O(\mathcal{C}) = 10^{-3}$ via belief propagation gauging after the application of each gate allows the simulation to reach a lower energy compared to not gauging. This is especially notable early in the evolution where we use a large step $\Delta\beta = 0.25$. Not enforcing the Vidal gauge condition causes the variational state to start growing in energy if too many steps are taken. The gauged state is significantly more robust to using large imaginary time-steps.

For the real time evolution in Fig. 6b we start in a Néel state and repeatedly apply layers of random two-site unitaries to the state. Each layer consists of a 'cross-hatch' of random unitaries, i.e. random gates are applied across the horizontal bonds of the lattice and then applied across the vertical bonds. We enforce a maximum bond dimension $\chi$ and denote the state reached after applying the $n$th gate $U_n$ as $|\psi_n\rangle$. After the application of each gate we

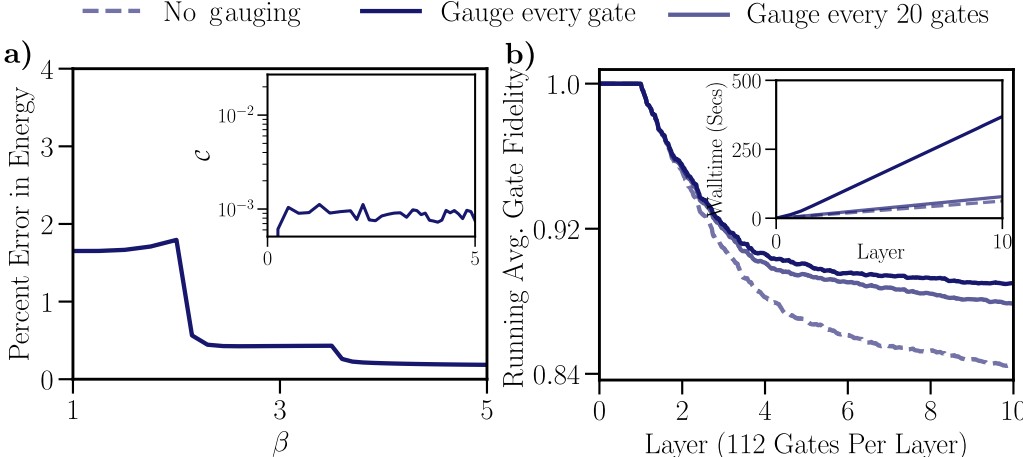

Figure 6: Imaginary and real time dynamics of a tensor network state on a 2D lattice, starting from an initial product state with neighboring spins polarized in opposite directions along the spin $z$ axis. Solid lines show evolution while performing belief propagation gauging after the application of a certain number of gates while the dashed line shows the case where no gauging is performed. **a)** Percent difference between energy of the 2D TNS during imaginary time evolution under $U = \exp(-\Delta\beta H)$ in Eq. (33) and energy found via DMRG on a MPS Ansatz with a bond dimension of 500. The Hamiltonian $H$ is the 2D transverse field Ising model with a field strength of $h = 3.0$. The lattice is size $10 \times 10$. Imaginary time steps of $\Delta\beta = 0.25, 0.15$, and 0.1 are used for 8, 10, and 15 applications of $U(\Delta\beta)$ respectively. The 2D TNS is limited to a maximum bond dimension of $\chi = 2$ and the energy is calculated via boundary MPS [114, 155] with a MPS bond dimension of $D = 10$; this is enough to achieve convergence in expectation values. For the solid line, the gauging routine targets $O(\mathcal{C}) = 10^{-3}$. Inset) Distance to Vidal gauge versus imaginary time. **b)** Average gate fidelity (Eq. (35)) for an $8 \times 8$ lattice and applying cross-hatched layers of random two-site unitaries with the bond dimension limited to $\chi = 4$. Fidelities are calculated using boundary MPS with a maximum dimension of $D = 10$. For the solid lines, the gauging routine targets $O(\mathcal{C}) = 10^{-3}$ and uses a serial update schedule for each belief propagation based on forest covers of the lattice and depth first search (see main text for further details). Each layer consists of 112 gates. Inset) Walltime, in seconds, of the different simulations.

use boundary MPS to compute the overlap of the approximate state and the state found if we applied the previous gate exactly, i.e. we compute

$$f_n = \left| \frac{\langle \psi_n | U_n | \psi_{n-1} \rangle}{\sqrt{\langle \psi_n | \psi_n \rangle \langle \psi_{n-1} | \psi_{n-1} \rangle}} \right|^2 . \tag{34}$$

We plot the running average gate fidelity

$$F(n) = \left( \prod_{i=1}^{n} f_i \right)^{\frac{1}{n}} , \tag{35}$$

a quantity which has been used recently to assess the fidelity of simulations of quantum circuits with tensor networks [28, 33, 34]. The initial state is always in the Vidal gauge with $\mathcal{C} = 0$. We observe that gauging after every 1 or 20 gate applications leads to a significant improvement in gate fidelity over the case of not gauging. Especially notable is that when gauging every 20

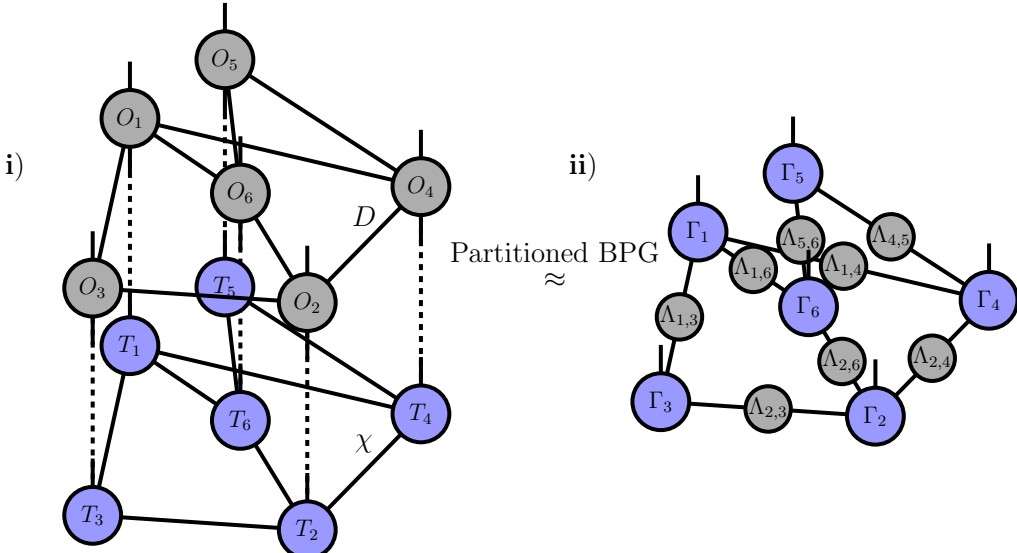

Figure 7: An example of BP gauging a partitioned network in the context of contracting a tensor network operator (TNO) composed of tensors $O_\nu$ with a tensor network state (TNS) composed of tensors $T_\nu$. **i)** - **ii)** The combined TNO-TNS network is partitioned, with the corresponding site tensors $O_\nu$ and $T_\nu$ grouped together. A generalized version of the belief propagation gauging routine is then performed on the partitioned structure and used to bring the state into the Vidal gauge and truncate the bonds of the resulting TNS to a desired dimension. The resulting state can then be transformed to the symmetric gauge via Eq. (27) and a new TNO can be applied. We emphasize that contracting the tensors within a partition does not need to be performed explicitly and for efficiency the two-layer network can be kept in memory while performing belief propagation on the norm network. This can lead to further efficiency improvements from BP gauging over eager and simple update gauging [120, 131] which effectively require pre-contracting the groups of tensors within partitions in the first iterations, leading to larger intermediate tensors in subsequent iterations of gauging.

gates the simulation takes only 50% longer than when not gauging and yet demonstrates a substantial lowering of the gate error.

We have shown that maintaining the Vidal gauge during gate evolution of a TNS can improve the accuracy of simple update gate evolution, while preserving the same computational scaling of simple update. Having faster gauging routines makes this more practical and therefore expands the use cases of this method in 2D TNS (PEPS) calculations, since regauging the network can significantly increase the runtime of simple update gate evolution if it is performed too often. It may only be necessary to gauge a region of the network surrounding where a gate or set of gates has been applied with simple update, similar to a technique used in Ref. [29] (though that reference uses a slightly different gauge). This is because, in general, a gate will have a minimal affect on the gauge of a TNS in regions far from where it is applied. This will be investigated in future work. In a recent paper, we show the utility of regauging a network during simple update gate evolution (using the BP gauging method proposed in this paper) for simulating the dynamics of the kicked Ising model on a heavy-hexagonal lattice [34], a system that was recently emulated on IBM's Eagle quantum processor [187].

### 3.5 Belief propagation gauging on a partitioned network: Application to contracting tensor network operators with states

It is natural to consider a simple extension of belief propagation where the corresponding graphical model is first partitioned — i.e. local degrees of freedom are grouped together — and messages are defined between the different regions. This has previously been referred to as 'block belief propagation' [150] and is also related to the well-established generalized belief propagation method [143, 144, 154], although without the partitions overlapping.

In the context of gauging a tensor network state, partitioning can be used to generalize the belief propagation gauging procedure outlined in Fig. 3: a site tensor $T_v$ would be replaced by a group of tensors associated with the vertex group $v$. Then, each $\mathcal{T}_v$ would correspond to that associated group of tensors and their conjugates. Message tensors are then defined between the tensor groups $\mathcal{T}_v$ and iterated as usual. As is the case for generalized BP, the message tensors would generically grow exponentially with the size of the partitions, although this could be circumvented by approximating the message tensors themselves as tensor networks (such as MPSs [150]) and not just single tensors. The resulting message tensors can then be used to gauge the original network to obtain a $\Gamma_v$ site tensor associated with each vertex group $v$ and a $\Lambda_e$ bond tensor associated with each edge $e$ of the partitioned TNS, which then (approximately, up to BP convergence and possible truncations) obey the standard isometry conditions defined in Eqs. (1) and (2).

An application of BP gauging generalized to partitioned TNS is approximately contracting a tensor network operator (TNO) with a TNS, resulting in a new TNS, under the BP approximation. An example of this is illustrated in Fig. 7. TNOs arise naturally from long-range gates, layers of 3D classical partition functions and from grouping layers of gates together when performing gate evolution of a TNS. This latter application can be used as an alternative to evolving a TNS gate-by-gate and can provide higher accuracy since fewer truncations are performed. One can view the application of the TNO onto the TNS using the BP approximation as a two-layer network that is first partitioned by grouping each site tensor $T_v$ of the TNS with its corresponding site tensor $O_v$ of the TNO. BP gauging can then be used to regauge and truncate the resulting partitioned network, similar to previous work that used eager gauging [131] and simple update gauging [120] for the same task. In contrast with those methods, however, BP gauging does not require contracting the corresponding sites tensors of the TNO with the TNS ahead of time, since BP can be run on the partitioned network structure [143, 150, 154]. This can give further efficiency improvements from BP gauging compared to eager and simple update gauging beyond those we already demonstrated in Sec. 3.2 for gauging simple (un-partitioned) TNS. Those methods effectively require pre-contracting the groups of tensors within partitions in the first gauging iteration, leading to larger intermediate tensors in subsequent iterations of gauging. BP gauging of partitioned tensor networks was used in Ref. [], where layers of gates of a 2D quantum circuit were represented as TNOs.

### 3.6 Gauging infinite tensor networks with belief propagation

So far, we have mostly focused our attention on gauging finite tensor networks. Infinite tensor network states are, however, an important branch of research on tensor networks as they allow computations which work directly in the thermodynamic limit. There is extensive literature on gauging infinite tensor network states [116, 120, 131, 133].

Here we demonstrate how our belief propagation gauging routine can be used to gauge an infinite TNS and to compute observables under the BP approximation. Specifically, consider an infinite TNS which can be constructed by repeated translation of a finite tensor network. We can use belief propagation gauging to determine the corresponding bond and site tensors which transform the infinite TNS into the Vidal gauge. Specifically, one can take the tensor

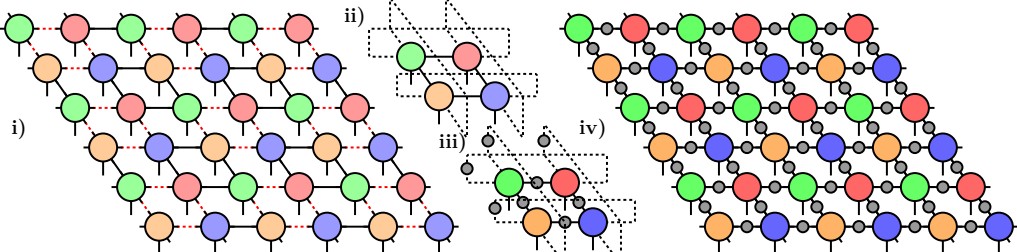

Figure 8: Gauging infinite tensor networks with BP gauging. **i)** Infinite 2D TNS (PEPS) with a $2 \times 2$ unit cell. Unit cells are separated by red dotted lines. **ii)** Unit cell of the infinite 2D TNS but with additional periodic edges added (dashed lines). **iii)** Upon gauging the periodic unit cell new site tensors and bond tensors are determined which put it into the Vidal gauge. The bond tensors are not necessarily identical, and can be unique for each edge (in this example there are 8). **iv)** The resulting site tensors and bond tensors from the periodic TNS determine the Vidal gauge for the infinite lattice. These tensors can be used to directly extract observables from the infinite TNS, for example via Eq. (36).

network state over the finite unit cell, add in appropriate periodic boundary conditions (in the form of additional bonds on the lattice), run BP on the periodic TNS and then use Eqs. (21), (22), and (23) to determine the corresponding bond tensors and site tensors which yield the periodic TNS in the Vidal gauge. It can be proven that these tensors are exactly those that transform the infinite TNS into the Vidal gauge. Fig. 8 illustrates gauging an infinite TNS for the square lattice with a $2 \times 2$ unit cell — although we emphasize our method is independent of the network structure and can be easily applied to unstructured unit cells in arbitrary dimensions [122].

In order to demonstrate this procedure computationally, we consider the TNS corresponding to the square root of the Ising partition function on an infinite square lattice. The site tensors $T_v$ are all identical, with $z = 4$, and their construction is described in Refs. [149,173]. As a representation of the infinite TNS we take the same tensors as the finite case but on the vertices of a $3 \times 3$ unit cell with periodic boundary conditions in both horizontal and vertical directions. We gauge this state with our belief propagation gauging routine and also gauge the same state but on a finite, open boundary $L \times L$ lattice.

In Fig. 9a, upon gauging the state, we approximately measure $\langle S_v^z \rangle$:

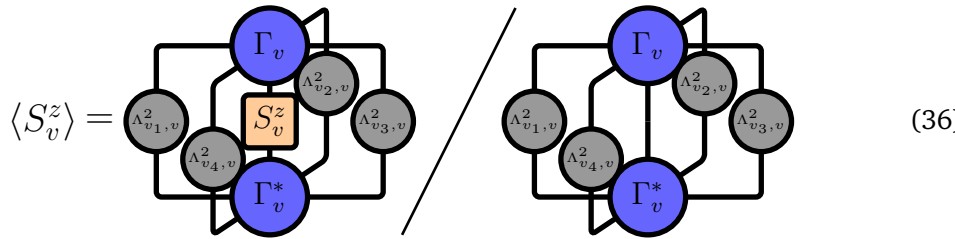

$$\langle S_v^z \rangle = \qquad\qquad\qquad\qquad / \qquad\qquad\qquad\qquad \tag{36}$$

which essentially gives the best approximation for $\langle S_v^z \rangle$ under the assumption of a rank-one environment. It is equivalent to running belief propagation on the norm network of the periodic

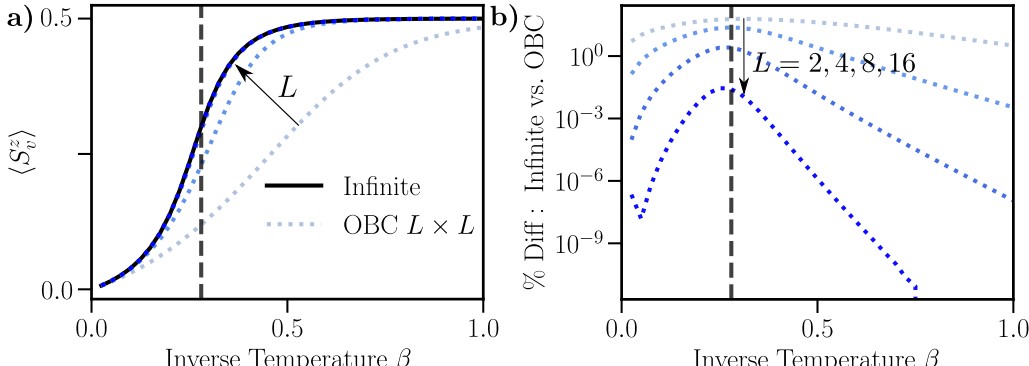

Figure 9: Gauging the tensor network state corresponding to the square root of the classical Ising partition function on a square lattice, with a longitudinal field of $h = 0.5$. Black dashed line represents the point $\beta_c \approx 0.28$, which is the critical point under the Bethe approximation. **a)** Result from gauging the state and approximately measuring $\langle S_v^z \rangle$ using Eq. (36), taking $v$ to be the central site of the lattice. The solid black line shows the result from a $3 \times 3$ periodic lattice (which is equivalent to an infinite lattice under the BP approximation — see text) while dotted lines show the result for an open boundary $L \times L$ lattice of increasing size ($L = 2, 4, 8$, and 16) with the result converging to the infinite one. **b)** Percentage difference between the infinite and open boundary values of $\langle S_v^z \rangle$ for increasing open boundary lattice size ($L = 2, 4, 8$, and 16).

unit cell and measuring $\langle S_v^z \rangle$ using the fixed point message tensors:

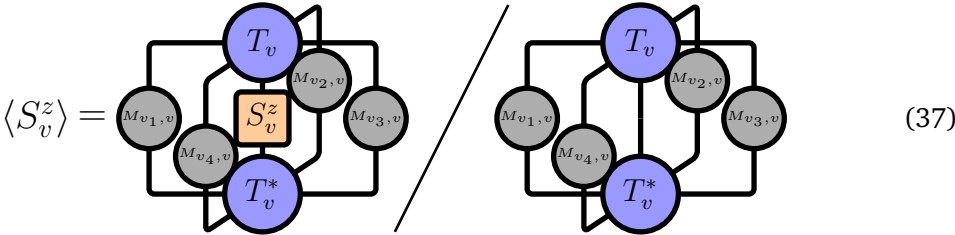

$$\langle S_v^z \rangle = \quad \Big/ \quad \tag{37}$$

The results in Fig. 9 show that the expectation value for $v$ in the middle of an increasingly large $L \times L$ open-boundary lattice converges to that of the small periodic unit cell. Using belief propagation to gauge the periodic TNS thus can be used to obtain results, under the approximation in Eq. (36), in $O(1)$ time in the lattice size for infinite networks (and $O(L)$ time where $L$ is the number of sites of the unit cell). This fact is implicitly used in the simple update gate evolution algorithm when applied to infinite PEPS.

## 4 Conclusion

In this work we have demonstrated how belief propagation (BP) can be used to bring an arbitrary tensor network state (TNS) into a gauge commonly used in the tensor network literature — which we refer to as the Vidal gauge but has previously been referred to as the super-orthogonal or quasi-canonical gauge — where multiple local orthogonality conditions are (approximately) satisfied. We showed that it can be viewed as a simplified version of a gauging method previously introduced in Refs. [131, 133]. We have benchmarked our 'belief propagation gauging' algorithm, demonstrating that it can be faster than existing algorithms for reaching the Vidal gauge. Additionally, we demonstrated the application of our new gaug-

ing algorithm to improving the accuracy of simple update gate evolution of TNS. We also discussed 'block BP gauging' which involves performing belief propagation gauging on a partition tensor network. We showed an example of block BP gauging in the context of the contraction of a TNS and a tensor network operator (TNO), and we argued that it has additional efficiency advantages in this context over currently available gauging methods because it can more effectively make use of a partitioned tensor network structure, which is utilized in the recent Ref. []. Finally, we demonstrated — by gauging small periodic systems — how to gauge infinite tensor network states with our algorithm and how this can be used to make approximate measurements corresponding to the thermodynamic limit.

Our new BP gauging method has a number of potential advantages over previous ones, in addition to the ones mentioned above. Because it is primarily based on BP, a well established method, optimized BP implementations as well as algorithmic advancements to BP (like the recent [151]) and generalizations of BP [143,150,154] can be repurposed for the application of gauging and truncating TNS. Additionally, the BP algorithm is very simple and is based only on tensor contractions (whereas previous gauging methods interspersed tensor contractions and factorizations). Therefore it is simpler to implement, optimize, and should be more straightforward to effectively make use of specialized hardware like GPUs or TPUs [189], where tensor contractions generally have bigger speedups than factorizations compared to CPUs. On a theoretical level, it would be interesting to investigate connections between various TNS gauging proposals (for example [125,127,132,137,138]) and explore if there is a more unified picture based on tensor contraction, in anology to the close connection that is now established between the belief propagation tensor network contraction algorithm and standard tensor network gauges.

## Acknowledgments

We would like to thank Miles Stoudenmire for illuminating discussions. We would also like to thank Miles Stoudenmire, Johnnie Gray, and Nicola Pancotti for providing helpful comments on the manuscript.

**Funding information**   J.T. and M.F. are grateful for ongoing support through the Flatiron Institute, a division of the Simons Foundation.

**Computing resources and software packages**   The code used to produce the numerical results in this paper was written using the **ITensorNetworks.jl** [169] package — a general purpose and publicly available Julia [190] package for manipulating (gauging, contracting, partitioning, evolving, truncating, optimizing, etc.) tensor network states of arbitrary geometry. It is built on top of **ITensors.jl** [191], which provides the basic tensor operations. Code is available in the current version of ITensorNetworks.jl for performing belief propagation gauging on arbitrary tensor network states. Code for using the other routines described in this paper is also available. An example is available which benchmarks these routines against each other. Our benchmarking was done using the Rusty cluster housed in the Flatiron Institute in New York, New York. All code was run using 10 cores of the same Skylake computing node.

The tensor network diagrams in this paper were produced using the publicly available package **GraphTikz.jl** [192], a general purpose Julia package for visualizing graphs, including tensor networks.

# A   Simple update

Here we detail more efficient variants of the the simple update (SU) procedure — whose most direct implementation is shown in Fig. 5 — for applying a two-site gate $O_{v,w}$ to a TNS [116, 119, 122, 139]. The original TNS is taken to be in the Vidal form although we emphasize that the same efficiency improvements directly apply to the analogous BP variant of simple update (see Fig. 5b). Methods — such as the 'reduced tensor' method [96, 119, 121, 132, 193] — have been developed to improve the efficiency of the SU procedure by using orthogonal decompositions like the QR decomposition to apply the gate to a reduced space. In this work, when applying a gate to a TNS, we utilize this more efficient version of SU [96, 193]. We depict this version diagrammatically in Fig. 10.

When performing simple update gauging, an identity gate is applied. This means the gate in Fig. 10 can be removed entirely and the physical, site indices can be moved onto the $Q$ tensors during the QR decomposition in order to improve efficiency [186]. This is the procedure we adopt when performing edge updates in the simple update gauging procedure (which is described in the following section). We illustrate this in Fig. 11.

# B   Gauging routines review

Here, we provide details of the existing gauging routines that we benchmark our 'belief propagation gauging' routine against in Fig. 4 of the main text.

The first, which we refer to as 'simple update gauging', involves repeated iterations of simple update (without any truncation) across the edges of a TNS with identity gates (see Fig. 11). This is known to bring the network closer to satisfying Eqs. (1) on every edge [120].

To be explicit, we implement the following routine to produce the results for 'simple update gauging' presented in Fig. 4 of the main text:

---

**The simple update gauging routine**

- Start with a TNS consisting of site tensors $\{\Gamma_v\}$ and bond tensors $\{\Lambda_e\}$. If the TNS is described only by site tensors (no bond tensors) then assign the bond tensors to identity matrices.

- Perform an iteration:

    - Iterate over each edge of the TNS and perform simple update with identity gates (i.e. the simple update variant shown in Fig. 11) to get new site tensors $\{\Gamma_v\}$ and bond tensors $\{\Lambda_e\}$.

- Repeatedly perform iterations until the designated stopping criteria is reached.

---

The second gauging routine we benchmark against we refer to as 'eager gauging', which was first introduced in Refs. [131, 133]. Specifically, the routine is equivalent to the following[8] and was used to produce the results for 'eager gauging' presented in Fig. 4 of the main text:

---

[8]We formulate it here in terms of the symmetric gauge which makes the comparison to BP gauging clearer. Our actual implementation for the purpose of benchmarking directly follows the steps outlined in Refs. [131, 133].

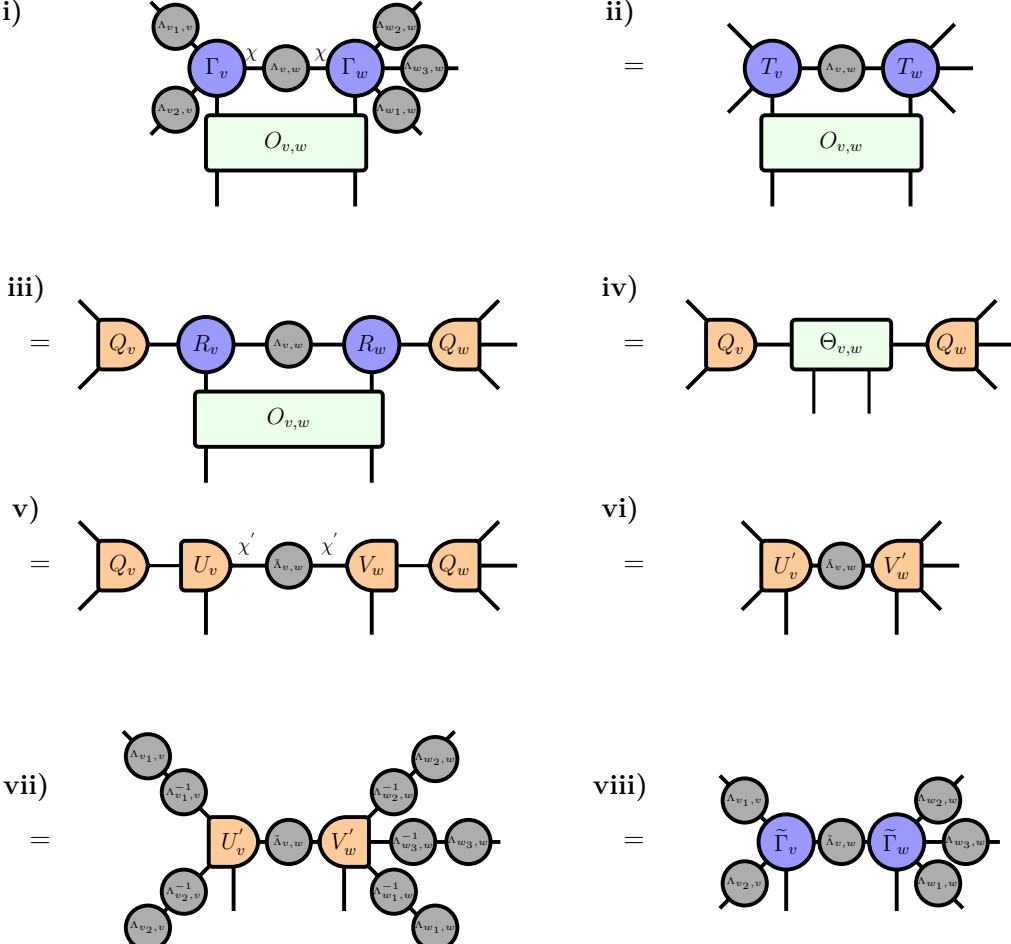

Figure 10: 'Reduced tensor' version of simple update described in Fig. 5. The example pictured shows a two-site gate $O_{v,w}$ being applied to neighbouring sites $v$ and $w$ with coordination numbers of 3 and 4 respectively. The dimension of the bond along the edge $e = (v,w)$ is $\chi$. **i) - ii)** The bond tensors are absorbed onto the site tensors. **ii) - iii)** A QR decomposition is performed on both the left and right site tensors. **iii) - iv)** The resulting $R_v$ and $R_w$ tensors, the bond tensor $\Lambda_{v,w}$, and the gate are combined together to form the composite tensor $\Theta_{v,w}$. **iv) - v)** The composite tensor is SVDd (with the bond being truncated down to dimension $\chi'$) and the resulting singular values become the new bond tensor $\tilde{\Lambda}_{v,w}$. **v) - vi)** The $U_v$ and $V_w$ matrices from the SVD are multiplied back with $Q_v$ and $Q_w$. **vi) - vii)** Resolutions of identity using the previous bond tensors are inserted. **vii) - viii)** The inverses are absorbed to yield the updated bond tensors and site tensors. Such a procedure could be generalized to long-range gates by matrix product operator decompositions of the gate.

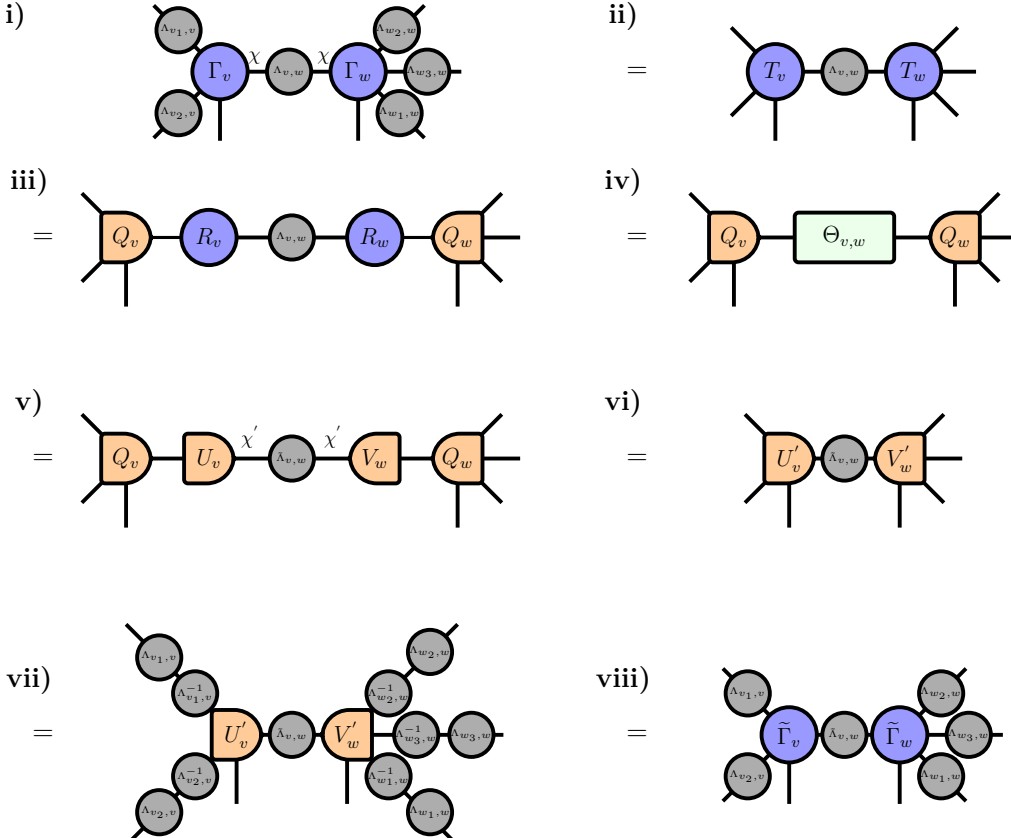

Figure 11: Simple update step when applying an identity matrix. This step is the workhorse of simple update gauging. The example pictured is two neighbouring sites with co-ordination numbers of 3 and 4 respectively. The bond dimension along the edge $e = (v, w)$ is $\chi$. **i) - ii)** The bond tensors are absorbed onto the site tensors. **ii) - iii)** A QR decomposition is performed on both the left and right site tensors. **iii) - iv)** The resulting $R$ tensors are combined together to form the composite $\Theta_{v,w}$. **iv) - v)** The composite tensor is SVDd (with the bond being truncated down to dimension $\chi' \leq \chi$) and the resulting singular values become the new bond tensor $\tilde{\Lambda}_{v,w}$. **v) - vi)** The $U_v$ and $V_w$ matrices from the SVD are multiplied back with $Q_v$ and $Q_w$. **vi) - vii)** Resolutions of identity using the previous bond tensors are inserted. **vii) - viii)** The inverses are absorbed to yield the updated bond tensors and site tensors.

**The eager gauging routine**

- Start with a TNS consisting of site tensors $\{T_v\}$. Form the closed, norm network of the TNS.

- Initialize the belief propagation message tensors of the norm network to arbitrary positive definite matrices — in our implementation we initialize them to identity matrices.

- Perform an iteration:

  - Do a single update of each message tensor via Eq. (5).
  - Gauge the TNS with the resulting message tensors via Eqs. (21), (22), and (23), yielding site tensors $\{\Gamma_v\}$ and bond tensors $\{\Lambda_e\}$.
  - Transform into the symmetric gauge by absorbing the square root bond tensors via Eq. (27), yielding a new set of site tensors $\{T_v\}$.
  - Assign the message tensors to the bond matrices found from gauging i.e. $M_e = M_{\bar{e}} = \Lambda_e$.

- Repeatedly perform iterations until the designated stopping criteria is reached, skipping the transformation into the symmetric gauge in the last iteration.

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
