# Peer review of "Gauging tensor networks with belief propagation"

_SciPost Physics, doi:SciPost Phys. 15, 222 (2023)_

## Round 2 · Referee Report · Anonymous (Referee 1) · 2023-9-30

Strengths

1- the intersection of tensor networks and tools such as BP is ripe for exploration. This paper both clarifies the links between BP and other algorithms such as Simple Update (SU), including the explicit gauge relation, and shows the practical advantage to using BP. 2- The paper is very clearly explained with diagrams and also structured, making it a good introductory and reference paper. 3- There is sufficient detail to reproduce everything, and the actual code is open source.

Weaknesses

1- Theoretically it was already known that BP and SU are somehow equivalent and should converge to the same thing, and thus a critical reader might argue that the techniques introduced here don't fundamentally change what simulations are accessible c.f. SU. 2- The actual results shown are all at quite small scale and show fairly mild improvements as compared to SU or no re-gauging. Again a critical reader might find these underwhelming, though my personal opinion is that BP will be indeed be very useful for large simulations, for various reasons including beyond pure performance.

Report

I think this is a very timely and clear paper, that makes another important link between BP and tensor networks, via the new gauging relation, and also shows the practical advantages of using. I think the paper is definitely worthy of publication, since, although one might argue there is not radically new theory or concepts, it provides a very clear, detailed and practical demonstration and how and why to use BP for tensor networks, a combination of methods that is likely to be very common in the future. I do think ideally the small points raised below should be addressed in some way.

One more minor comment for the authors: 1. the gauging procedure is the same as the projectors inserted in CTMRG and HOTRG, I don't know if this is a useful link for the authors to make or think about, or is maybe too trivial.

Requested changes

1- For the comparison with SU / eager BP and BP. As far as I understand for BP implemented here a 'parallel update' is always used (i.e. messages at step $t$ depend only on messages at step $t -1$). Whereas SU can be thought of as a sequential update. One is left wondering whether the difference in convergence is simply this difference. It might be nice to check with a 'sequential BP', where updated messages immediately replace the old ones. I think this is different from the eager BP shown - there is no gauging between steps required. While there are many convergence BP techniques and for one thing this might speed things up, more interestingly/importantly it might show whether BP and SU really converge at exactly the same rate (in terms of iterations) using the same update schedule. However, I would consider exploring this fully not strictly required.

2- The authors briefly discuss generalized BP and dub 'generalized BP gauging'. Though I am not an expert, I think what the authors refer to, including the referenced tensor network papers, is really still just BP but with larger 'regions'. Whereas a generalized BP algorithm (beyond the most simple case of BP itself) actually involves overlapping regions and a considerably more complicated relationship between the marginals and messages being passed around. As such, the name 'generalized BP gauging' might be a bit premature.

  • validity: top
  • significance: high
  • originality: good
  • clarity: top
  • formatting: perfect
  • grammar: perfect

Author:  Joseph Tindall  on 2023-11-09  [id 4103]

(in reply to Report 1 on 2023-09-30)

We thank the referee for their thorough review of our work. Especially for pointing out the importance of scheduling. The changes we have made in light of this have improved our work and the runtime of our algorithm.

Response to requested changes:

1) The referee is correct that the update schedule of the BP and SU affects their convergence and runtimes. This is an important point to address in order to make as unbiased comparison as possible and we are grateful to the referee for bringing this to our attention. Hence, for the benchmarking of the algorithms in Figure 4 we have now opted to fix a hybrid `parallel’/’sequential’ schedule to the updates based on a Trotterisation of the edges of the lattice into groups where edges in the same group share no common vertices. Updates on edges in the same group are performed in parallel but updates between groups are performed sequentially. Ideally, we would perform a fully parallel update to compare each algorithm but simple update gauging will not converge unless the edges which share common vertices are updated sequentially.

We find that with this fixed update schedule all three algorithms converge in the same number of iterations. Belief propagation gauging is still noticeably faster per iteration, and therefore also has a faster total runtime, due to the lack of extraneous operations per iteration. We have also added, to Fig. 4, a plot comparing the timings from belief propagation gauging with various schedules: fully parallel, the hybrid schedule defined above, and a custom sequential schedule we find to be most effective. We use this latter schedule in Figure 6 now which has improved the timings significantly. We have also referenced other update schedules for BP used in previous literature whose study could be the topic of future research.

2) We thank the referee for pointing out the fact generalized BP allows for overlapping regions and thus defines a hierarchy of messages between regions and sub-regions. To avoid confusion we have clarified that we do not consider overlapping regions when partitioning the tensor network. We have also avoided dubbing the algorithm generalized BP gauging and instead refer to it as `BP gauging on a partitioned network’. We have included the relevant citations.

---

## Round 2 · Referee Report · Anonymous (Referee 2) · 2023-10-11

Strengths

1- very clearly written, with a lot of background information and comprehensive list of references; the text is also a good review work for gauge fixing in tensor networks; 2- new algorithm for gauge fixing a general tensor network, providing an approximate local environment for each tensor; 3- benchmarks showing good performance in different contexts.

Weaknesses

No apparent weaknesses.

Report

In this work, the authors introduce a new algorithm for finding an effective gauge of a general tensor networks. For the case of loop-free tensor networks (matrix product states and tree tensor networks), this algorithm reduces to the standard canonical form; for "loopy" tensor networks this algorithm converges to what the authors call the "Vidal gauge". The algorithm is explained in good detail and carefully benchmarked.

As the authors show, this gauge is often used implicitly in simple-update-style algorithms, which is often quite useful, but also often fails in capturing the environment accurately. The authors clearly discuss this drawback and make it clear that these types of algorithms should be used with this in mind.

Therefore, this is an interesting work that pushes the efficiency of simple-update-style algorithms, and is therefore expected to be commonly used in situations with only tree-like quantum correlations.

I recommend publication without any requested changes.

Requested changes

I don't request any changes.

  • validity: high
  • significance: good
  • originality: ok
  • clarity: top
  • formatting: perfect
  • grammar: perfect

Author:  Joseph Tindall  on 2023-11-09  [id 4102]

(in reply to Report 2 on 2023-10-11)

We thank the referee for their thorough review of our work and positive comments.

---

## Round 3 · Author Response

Dear Editor and Referees,

We thank the referees for their reviews and thorough comments on our article 'Gauging tensor networks with belief propagation’. The referees found our work 'timely’ and 'worthy of publication’. Referee 2 directly recommended publication with no changes whilst Referee 1 has several requested changes which we have addressed and responded to.

We hope that the manuscript is now fit for publication in SciPost Physics.

The authors,

Joseph Tindall and Matt Fishman

---

## Round 3 · List of Changes

1) We have updated Figure 4 to fix the update schedule when comparing algorithms. We have also changed Fig. 4d to a benchmark the timings of belief propagation gauging using various update schedules.
2) We have discussed, in section 3.2, the importance of update schedules and its effect on the runtime of the algorithms. We have provided extensive references on the subject of scheduling in belief propagation.
3) We have changed the value of O(C) targeted in Fig. 6b to $10^{ -6}$ from $10^{-3}$. We have also changed the update schedule for BP to a sequential one based on a custom sequence. This has significantly improved timings whilst maintaining the same fidelity.
4) We have avoided the term 'generalized BP gauging’ and instead referred to it as 'BP gauging on a partitioned network'. We have also pointed out explicitly that this method does not consider overlapping partitions, which is possible in 'generalized BP' as defined in the literature.
5) We have added a section title 'Using square root belief propagation to gauge a tensor network state' to the discussion on square root belief propagation.
6) Minor grammatical changes throughout.

---

## Editorial Decision

published